# New Dawn for Atherosclerosis: Vascular Endothelial Cell Senescence and Death

**DOI:** 10.3390/ijms242015160

**Published:** 2023-10-13

**Authors:** Lan-Lan Bu, Huan-Huan Yuan, Ling-Li Xie, Min-Hua Guo, Duan-Fang Liao, Xi-Long Zheng

**Affiliations:** 1School of Pharmacy, Hunan University of Chinese Medicine, Changsha 410208, China; lanlan_bu@hnucm.edu.cn (L.-L.B.); dfliao@hnucm.edu.cn (D.-F.L.); 2College of Integrated Chinese and Western Medicine, Hunan University of Chinese Medicine, Changsha 410208, China; 20213749@stu.hnucm.edu.cn (H.-H.Y.); lingli.xie@ucalgary.ca (L.-L.X.); 20223820@stu.hnucm.edu.cn (M.-H.G.); 3Departments of Biochemistry and Molecular Biology and Physiology and Pharmacology, Cumming School of Medicine, University of Calgary, Calgary, AB T2N 1N4, Canada

**Keywords:** atherosclerosis, endothelial cell, endothelial cell senescence, endothelial cell death

## Abstract

Endothelial cells (ECs) form the inner linings of blood vessels, and are directly exposed to endogenous hazard signals and metabolites in the circulatory system. The senescence and death of ECs are not only adverse outcomes, but also causal contributors to endothelial dysfunction, an early risk marker of atherosclerosis. The pathophysiological process of EC senescence involves both structural and functional changes and has been linked to various factors, including oxidative stress, dysregulated cell cycle, hyperuricemia, vascular inflammation, and aberrant metabolite sensing and signaling. Multiple forms of EC death have been documented in atherosclerosis, including autophagic cell death, apoptosis, pyroptosis, NETosis, necroptosis, and ferroptosis. Despite this, the molecular mechanisms underlying EC senescence or death in atherogenesis are not fully understood. To provide a comprehensive update on the subject, this review examines the historic and latest findings on the molecular mechanisms and functional alterations associated with EC senescence and death in different stages of atherosclerosis.

## 1. Introduction

Atherosclerosis is a chronic cardiovascular disease (CVD) that poses significant risks to human health, and is the underlying cause of peripheral vascular disease, coronary heart disease, and stroke. The pathogenesis of atherosclerosis is complex and involves various cell types, including endothelial cells (ECs), vascular smooth muscle cells (SMCs), adventitial fibroblasts, macrophages, and other immune cells. Key factors in the development of atherosclerosis include endothelial dysfunction, leukocyte adhesion, foam macrophage formation, and SMC phenotypic transition. Endothelial dysfunction is considered the initial step in atherosclerosis, and in its broadest sense, it encompasses a constellation of nonadaptive alterations in functional phenotype, which have important implications for the regulation of hemostasis and thrombosis, local vascular tone, redox balance, and the orchestration of acute and chronic inflammatory reactions within the arterial wall [1]. ECs that line elastic arteries, such as the aorta, carotid artery, and femoral artery, have critical functions in maintaining vascular homeostasis. The primary function of the endothelium is to produce nitric oxide (NO) and other vasoactive substances, including hydrogen sulfide (H_2_S), carbon monoxide (CO), endothelial-derived hyperpolarizing factor (EDHF), prostacyclin, endothelin, and hydrogen peroxide (H_2_O_2_), to regulate vascular tone. ECs form a continuous monolayer barrier that controls substance exchange among the lumen, vascular wall, and parenchyma. A specialized barrier function of the endothelium involves its immunoregulatory effects on leukocyte recruitment. Quiescent endothelium is immunosuppressive, with a surface glycoprotein profile that prevents leukocyte adhesion, crawling, and extravasation. Upon tissue injury and inflammatory stress, activated ECs present adhesive molecules, such as vascular cell adhesion molecule-1 (VCAM-1), intercellular adhesion molecule-1 (ICAM-1), and E-selection, to the cell surface to facilitate the transendothelial migration of leukocytes. The reactive, pro-inflammatory phenotype of ECs is indispensable for tissue repair after acute injury. However, in the context of chronic tissue damage, such as atherosclerosis, persistent endothelial inflammation becomes pathogenic. Moreover, the regenerative capacity of ECs is intrinsically critical to the re-endothelialization of the surface-eroded arterial lumen and the stabilization of atherosclerotic lesions. Notably, most of these endothelial dysfunctions are associated with endothelial cell senescence and death.

Cellular senescence is a process in which cells undergo permanent cell cycle arrest, with an altered secretome to remodel neighboring cells and the extracellular matrix (ECM) microenvironment [2]. Notably, vascular aging in animal models and humans is characterized by impaired endothelium-dependent dilation (EDD) [3], perturbed fibrinolysis [4], enhanced permeability [5], and aberrant angiogenesis [6]. In humans, endothelium-dependent vasodilation, usually measured as flow-mediated dilation of the radial artery, serves as a non-invasive marker of vascular aging and cardiovascular damage, even in the absence of clinical symptoms [7]. Importantly, cellular senescence of the endothelium is an integral component of vascular aging, as well as atherosclerosis [8]. EC senescence triggers structural and functional deterioration of the vascular wall by not only deterring re-endothelialization and barrier reconstitution at the injury zone, but also promoting an inflammatory and thrombotic niche via the senescence secretome, thereby contributing to the development and progression of CVD [9]. Similar to senescence, multiple forms of endothelial cell death have been associated with the progression of atherosclerosis and the stability of plaques [10]. The death of ECs not only directly causes vessel denudation, but also impacts the neighboring environment via damage-associated molecular patterns (DAMPs) from dying or dead cells. Therefore, investigating the functional impacts of endothelial cell senescence and death will enhance our understanding of the pathogenesis of atherosclerosis and provide theoretical frameworks to develop more effective therapeutic strategies.

## 2. Vascular Aging and EC Senescence in Atherosclerosis

### 2.1. What Is Vascular Aging?

#### 2.1.1. Vascular Aging

Vascular aging is a specialized type of aging process involving the structural and functional deterioration of the vascular system. The clinically assessable features of vascular aging include reduced endothelium-dependent dilation and increased arterial stiffness, pulse wave velocity, systolic blood pressure, and central venous pressure, involving a spectrum of perturbed behaviors of ECs, SMCs, infiltrated immune cells, and the ECM. Importantly, vascular aging is a predictor of degenerative vascular diseases, including hypertension, aortic aneurysm, and atherosclerosis [11].

Aged ECs lose their balance in the production of vasodilatory and vasoconstrictive agents, characterized by decreased vasodilatory signals, such as NO and prostacyclin, and increased vasoconstrictive signals, such as endothelin and reactive oxygen species (ROS). Aged SMCs have perturbed mechanosensing and compromised ECM-remodeling capacity, which causes adverse remodeling of elastin and collagen in large elastic arteries. The fragmentation and degradation of elastin occur in the arterial media, driven by the upregulation of matrix metalloproteinases (MMPs) from SMCs and infiltrated leukocytes. The deposition of collagen, potentially as a mechanical compensation for elastin loss, accelerates advanced glycation end product formation, in turn promoting the cross-linking of structural proteins, exacerbating arterial stiffening, and compromising arterial compliance. Moreover, reduced proliferative potential, chronic low-grade inflammation, and non-permissive mechanics act synergistically to impair the endogenous repair capacities of ECs and SMCs so as to counteract age-related “wear and tear” damage.

#### 2.1.2. Morphological, Biochemical, Mechanical, and Metabolic Features of Senescent ECs

At the molecular level, cellular senescence, as an integral process of cellular aging, reflects a progressive failure of molecular mechanisms that leads to disturbances in DNA and cell nuclear and cytosolic environment [12]. The concept of cellular senescence, initially proposed in 1965 by Hayflick, suggests that it results from telomere shortening due to problems with the replication of evolutionarily conserved telomere ends. Subsequent studies identified cellular senescence as a permanent state of cell cycle arrest.

Senescent cells, including ECs, exhibit enlarged and flattened cell bodies. Such morphological features are closely associated with an aberrant hyper-adhesive phenotype [13]. Cho et al. reported hyperactivated focal adhesion kinase (FAK) and Rho GTPases Rac1, Cdc42, resulting in a larger focal adhesion complex and reduced cell motility [13]. Furthermore, multiple kinase pathways, such as mitogen-activated protein kinase (MAPK), mammalian target of rapamycin (mTOR), and Src, have been reported to play roles in senescent morphology [14,15]. Consequently, senescence alters the cell mechanics [16,17,18,19]. For example, Brauer et al. used traction force microscopy to demonstrate the increase in both adhesive and contractile force of senescent human skin fibroblasts at a single-cell resolution [16]. Such cellular biomechanical changes translate into the stiffening and contraction of an ex vivo 3D fibroblast-collagen construct, with a shifted capacity for collagenous matrix remodeling [16]. Most importantly, the morphological–mechanical coupling of senescent ECs has recently been described. Chala et al., with FluidFM-based single-cell force spectroscopy, reported that senescence of ECs leads to an enlarged spread area, enhanced EC-substratum adhesion, and loss of the ability to adapt their shapes in response to flow [20]. However, it must be noted that most morphological and biomechanical features of senescence are characterized in cultured cells. This warrants further study on the in vivo morphology and mechanics of EC senescence in aged or diseased vascular walls.

Intriguingly, senescent cells are metabolically active, albeit with dysregulated profiles. A wide spectrum of essential cellular metabolic processes is impacted by senescence [21,22], including in ECs [22,23]. Importantly, many senescence-related metabolic processes of ECs, including glycolysis, oxidative phosphorylation, fatty acid oxidation, and NAD^+^ metabolism, have been indicated for their functional involvement in vascular aging and/or atherosclerosis [22,23,24]. One particularly notable, but not yet explored process is glutamine metabolism. Specifically, glutamine metabolism in ECs maintains eNOS activity, suppresses inflammatory signaling, and, notably, plays an anti-senescent role in ECs [25]. The pharmacological inhibition of glutaminolysis, a process that converts glutamine into glutamate and ammonia, accelerates the replicative senescence of cultured ECs [26]. Moreover, deprivation of glutamine from the culture medium, or inhibition of glutaminase 1 (GLS1), a key glutaminolysis enzyme, in cultured ECs causes cell cycle arrest at the G0/G1 phase and sensitizes cells to cytotoxicity by H_2_O_2_ [27]. To date, the in vivo link between EC glutaminolysis and senescence in the context of atherosclerosis remains to be explored, although a pathogenic role of excess glutaminolysis in ECs and SMCs has been reported in human and rodent pulmonary hypertension [28], where EC senescence has a disease-driving role [29].

In addition to cell-autonomous effects, one critical outcome of cellular senescence is the modification of the cellular secretome, termed the senescence-associated secretory profile (SASP), which has been recognized as a crucial factor in age-related CVD, including atherosclerosis, and will be discussed in detail in the following sections.

A plethora of molecular mechanisms underlie senescence, including oxidative stress, inflammation, genomic instability, metabolite sensing longevity pathways (sirtuins, AMP-activated protein kinase/AMPK, mTOR, Klotho, fibroblast growth factor 21/FGF21), the activated renin angiotensin–aldosterone system (RAAS), uric acid and polyploidy which are the main topics of this review [30,31].

Currently, clinical assessments of aging are mostly made according to chronologic age, but natural aging does not assume a simple linear rate. The acceleration of organismal aging occurs around the last quarter of the lifespan, when senescent cells accumulate [32,33]. Critically, the rate of cellular senescence varies in the same tissue among different individuals, and among different tissues in the same individual [34,35]. To quantitatively assess the extent of cellular senescence in vivo, multiple methods have been developed, although a clinical assay has not yet been established. Specific advantages and considerations of each senescence assay are summarized in Table 1.

#### 2.1.3. Cellular Senescence in Atherosclerosis

Atherosclerosis has a complex pathogenic course that involves multiple cell types, including ECs, SMCs, fibroblasts, perivascular adipocytes, and immune cells. Here, we briefly discuss the evidence of senescence of these cell types in the context of atherosclerosis. Senescent ECs have reduced NO production and compromised barrier functions attributed to disorganization intercellular adherens junctions and tight junctions [50]. The endothelial coverage of atherosclerotic lesions is one critical protective factor for plaque stability, and attrition of the endothelium leads to atherothrombosis at the affected lesion surface [51,52]. Moreover, senescent ECs have a heightened ability to attract leukocytes via increased expression of adhesive molecules such as VCAM-1 and ICAM-1. The heightened endothelial inflammation is considered as a detrimental process that maintains the chronic inflammatory environment of the atherosclerotic wall.

SMCs are the major cellular origin of atherosclerotic lesion. Lineage-tracing studies have demonstrated that up to 50% of lesional cells, including lipid-laden foam cells, are derived from pre-existing medial SMCs. SMC senescence is considered to be a major disease-driving force of atherogenesis, as senescent SMCs not only produce inflammatory cytokines such as tumor necrosis factor α (TNFα), MCP-1, and IL-6, but also express matrix-degrading enzymes such as MMP9 [53]. These complex senescence-associated dysfunctions of SMCs lead to reduced fibrous cap thickness and increased necrotic core size, which are features of plaque instability [54].

Macrophages have a plethora of pro- and anti-atherogenic functions. Once they have infiltrated into the arterial wall, macrophages can take up lipids/cholesterol and convert into foam cells, constituting the cellular lipid storage of the lesion. Notably, the relative contribution to foam cells by macrophages and SMCs is still a matter of active debate. Macrophages can undergo phenotypic polarization. While M1-polarized macrophage amplify atherosclerosis via the production of inflammatory cytokines, M2 macrophage can play a protective role [55]. Importantly, p16^INK4a^-induced cellular senescence promotes macrophage differentiation into M1-type macrophages [56,57]. In addition, the senescence of T cells, B cells, and dendritic cells also makes significant contributions to vascular aging and atherosclerosis [57].

Perivascular adipocytes have been reported to support the formation of vascular injury-induced neointima, an early event in atherogenesis [58]. Furthermore, inflammatory activation, as assessed by histological examination, of perivascular adipocytes was reported with an atherogenic diet, suggesting an active role in atherosclerotic vascular wall remodeling [59]. However, in situ evidence for the senescence of perivascular adipocytes is currently lacking.

### 2.2. Mechanism of Endothelial Cell Senescence

ECs maintain homeostasis mainly by producing and releasing the vasodilator NO and the vasoconstrictor endothelin-1 (ET-1). NO is one of the most important EC-derived anti-atherogenic molecules; it maintains vasorelaxation and prevents arterial stiffening [60]. Aging leads to decreased NO bioavailability [61], tipping the balance to favor vasoconstriction caused by ET-1 or other vasoactive stimuli (ROS, angiotensin II (Ang II), etc.), thereby contributing to age-related decline in vasoreactivity and end-organ perfusion [62]. Importantly, aged ECs display elevated expression of ET-1 and synthesize prostanoids, which in turn enhances vasodilation [63]. Notably, endothelial dysfunction is not only a product of aging and senescence, but also a reciprocal contributory factor in vascular senescence. Inactivated endothelial nitric oxide synthase (eNOS) and increased eNOS uncoupling in ECs enables the production of ROS or reactive nitrogen species (RNS) instead of NO, which disseminate prosenescent signals into the vascular wall [64]. Moreover, an aged or senescent endothelium has an elevated expression of adhesive molecules, such as VCAM-1, ICAM-1, and E-selectin, that attract leukocytes, thereby contributing to chronic vascular wall inflammation [7,65]. In the following sections, we will discuss the triggering of and regulatory signaling for endothelial senescence, including oxidative stress, inflammation, genomic instability, metabolite-sensing longevity pathways (sirtuins, AMPK, mTOR, Klotho, and FGF21), activated RAAS, uric acid, and polyploidization.

#### 2.2.1. Oxidative Stress

Senescent ECs produce high levels of ROS, mainly including superoxide, H_2_O_2_, hydroxyl radical, and peroxynitrite, which hinder the vasodilator activity of NO. The redox imbalance is the main cause of endothelial dysfunction. ROS, at normal physiological levels, act as signaling molecules to regulate cell signaling, growth, and differentiation [66]. However, ROS at pathological levels result in oxidative stress, cell dysfunction, senescence, and even cell death [67]. Primary sources of endothelial ROS include mitochondria, NADPH oxidases, uncoupled eNOS, and xanthine oxidase (XO). Vascular stress signals, such as inflammation, Ang II, high glucose, oxidized low density lipoprotein (oxLDL), or even ROS itself, can serve as triggers of ROS production [68].

Mitochondria are the main ATP-producing organelle, and act as a primary source of ROS via the electron transport chain (ETC) [69]. Superoxide anion is produced by mitochondrial complex I and III via electron leakage from the ETC. Mitochondrial ROS (mtROS) further damages ETC complexes, in turn exacerbating ROS production [68]. Senescent ECs exhibit mitochondrial dysfunction, marked by a reduction in mitochondrial mass and changes in mitochondrial composition and ETC function. Aging is associated with reduced activities of complex IV and cytochrome c oxidase, both components of the mitochondrial ETC, leading to increased mitochondrial oxidative stress [70].

NOX4 is a subunit of the NADPH oxidase complex, which is arguably the most robust source of vascular ROS during atherogenesis. NOX4 can be activated by Ang II to generate superoxide, which reacts with NO to form peroxynitrite. NADPH-derived ROS and RNS, when present in excess, adversely impact endothelial biology, including endothelial inflammation, ischemic responses, and even NO production [71]. Specifically, Park and Datla et al. showed that the siRNA-mediated knockdown of NOX4 in human aortic or microvascular ECs significantly reduced ROS production and subsequent cell proliferation [72,73]. Furthermore, NOX4 downregulation inhibited endothelial inflammatory responses evoked by ROS, including ICAM-1, interleukin-8 (IL-8), and monocyte chemoattractant protein-1 (MCP-1) [72,73]. Importantly, NOX4 restricts the replicative lifespan of cultured human umbilical vein endothelial cells (HUVECs), and shRNA-mediated NOX4 knockdown delays the onset of replicative senescence [74]. Furthermore, downregulating NOX4 in ECs effectively inhibits replicative senescence by reducing DNA damage, which occurs independently of telomere attrition. The underlying mechanism by which NOX4 promotes senescence in ECs is associated with mitochondrial dysfunction [75]. On the other hand, NOX4 also has protective effects on vascular health. Mice with *Nox4* knockout (KO) have been shown to exhibit intensified Ang II-mediated inflammation, EC dysfunction, and vascular wall remodeling [76]. Furthermore, mice with NOX4 overexpression showed enhanced vasodilation and reduced blood pressure by producing H_2_O_2_ [77]. These dual roles of NOX4 in EC functions are consistent with the dose-dependent effects of ROS in vascular functions [78,79,80].

eNOS uncoupling occurs under the conditions of insufficient substrates and/or cofactors such as L-arginine or tetrahydrobiopterin (BH4). Uncoupled eNOS mainly generates superoxide, which reacts with NO to generate peroxynitrite. Notably, peroxynitrite is a highly reactive, toxic metabolite, not only oxidizing nucleic acids, proteins, and lipids, but also oxidizing BH4 to further deplete NO production and accelerate vascular aging [81,82,83,84]. Although peroxynitrite induces the senescence and death of human red blood cells [85], there has yet to be a direct report on the cause–effect relationship between peroxynitrite and senescence in ECs. In this regard, it is important to note that NO can inhibit EC senescence at baseline and at high glucose levels, contributing to the anti-aging actions of estrogen [86]. Under normal physiological conditions, inducible nitric oxide synthase (iNOS) is undetectable in ECs, but abundantly expressed when stimulated by endotoxin or cytokines [87]. Both eNOS and iNOS produce NO, but mainly in the endothelium and intima, respectively [88,89,90,91]. Moreover, the downregulation of eNOS activity and concomitant upregulation of iNOS activity in the arterial wall are prominent molecular features of vascular aging [88,89,90,91]. Notably, selective inhibition of iNOS can partially reverse senescence-related endothelial dysfunction [87].

XO produces ROS during the oxidative conversion of hypoxanthine into uric acid, serving as a main source of ROS during tissue inflammation and ischemic injury [92]. Notably, both ROS and uric acid can induce EC senescence, and the prosenescent role of uric acid will be discussed in the following section. Hypoxia was shown to increase the abundance and activity of XO in the serum and lungs in a neonatal rat model of hypoxic pulmonary hypertension with chronic exposure to 13% O_2_ for 14 days after birth. Intraperitoneal administration of allopurinol, an XO inhibitor, ameliorated oxidative stress in the lungs and adverse pulmonary vascular remodeling [93]. Although the roles of XO-derived ROS in EC senescence have yet to be reported, accumulating evidence has revealed a close relation between XO, EC dysfunction, and atherosclerosis. For example, when exposed to atheroprone oscillatory shear stress, bovine aortic ECs showed an increase in superoxide production, mediated by NADPH oxidase-dependent XO activation [94]. The administration of oxypurinol and allopurinol improved flow-mediated endothelial-dependent vasodilation of radial arteries in nine patients with coronary artery disease [95]. Recently, the PRIZE study, a multi-center randomized trial involving 514 patients with carotid atherosclerosis demonstrated that febuxostat, an XO inhibitor, can improve arterial stiffness over 24 months, albeit without significant benefits to the atherosclerotic lesion burden [96].

ROS is an active inducer of genomic instability [97]. Unrepaired DNA lesions, especially those that occur on telomeric DNA segments, can cause robust DNA damage responses (DDRs) [98]. Persistent DDR is one of the leading signaling triggers of senescence in many cell types, including ECs [99]. In this regard, it is particularly important to note the well-documented relationship between vascular cells, vascular aging, and atherosclerosis. This topic will be discussed in the following section. In addition to DNA damage, ROS can activate a variety of intracellular signaling pathways, such as NFE2L2-Kelch-like ECH-associated protein 1 (Nrf2-Keap1) [100], peroxisome proliferator-activated receptor gamma (PPAR-γ) [101], sirtuin 1 (SIRT1) [102], nuclear factor kappa-light-chain-enhancer of activated B cells (NF-κB) [103], and Forkhead box (FOXO) [104], which in turn modulate EC senescence [105,106,107,108]. Intriguingly, ROS-activated signaling cascades also frequently elevate the expression of antioxidant genes, including superoxide dismutase (SOD) [109], catalase [110], glutathione peroxidase [111], peroxiredoxins [112], and thioredoxin [113], likely as a compensatory feedback mechanism. Taken together, ROS, when present in excess, reduce NO bioavailability, impair endothelium-dependent vasodilation, and drive EC senescence, all of which contribute to vascular aging and atherogenesis. Oxidative stress and EC senescence are summarized in Figure 1.

#### 2.2.2. Inflammation

Chronic, low-grade inflammation is a key change that occurs in aging arteries [65]. Inflammation is argued to function as a link between the aging process and age-related diseases [114]. Compared with young people, the elderly have significantly increased plasma inflammatory markers associated with vascular diseases such as atherosclerosis, including TNF-α, interleukin-1β (IL-1β), interleukin-6 (IL-6), C-reactive protein (CRP), interferon γ (IFN-γ), MCP-1, and MMPs [115]. Such increases are proportional to age and independent of other cardiovascular risk factors [116]. Importantly, inflammation directly causes endothelial dysfunction and senescence, which constitute integral aspects of atherogenesis [117].

Transcription factor NF-κB has been established as the master regulator of inflammatory molecules, including TNF-α, interleukins (IL-1β, IL-2 and IL-6), chemokines (IL-8 and Regulation of Activation, Expression, and Secretion by Normal T Cells (RANTES)), adhesion molecules (ICAM-1 and VCAM-1), and enzymes (iNOS and COX-2) [118]. Thus, targeting the inhibition of these molecules could be a powerful strategy to prevent vascular aging and atherosclerosis. The recent CANTOS trial demonstrated that canakinumab, an IL-1β-targeting antibody, reduces the recurrence of cardiovascular events in patients with a previous myocardial infarction [119]. However, different types of vascular walls and circulating cells can contribute to local inflammation and atherosclerosis, but their relative contributions remain incompletely understood. Regarding endothelial inflammation specifically, Walker and Lesniewski et al. showed that inhibition of the NF-κB pathway significantly reduces inflammatory cytokine expression and enhances endothelium-dependent vasodilation in both aged mice and humans [120,121].

Senescence of vascular cells, including ECs, SMCs, and macrophages has been observed in atherosclerotic lesions [37]. Importantly, the elimination of p16^Ink4a^+ senescent cells alleviates atherogenesis and stabilizes established plaques in p16-3MR mice harboring a p16^Ink4a^ promoter-based apoptosis executioner cassette, in which cellular apoptosis is activated by exogenous ganciclovir [122]. Specifically, EC senescence plays causal roles in the progression of atherosclerosis. Telomeric repeat binding factor 2 (TRF2) is a shelterin complex protein that maintains the structural integrity of telomeres [123]. Cellular expression of a dominant negative mutant of TRF2 (TRF2 DN) causes the uncapping of telomeres and, consequently, telomeric DNA damage and senescence [54,124]. Honda et al. reported that mice with forced EC senescence, via Tie2-based expression of TRF2 DN, exhibited an increased burden of atherosclerotic lesions in apolipoprotein E (*Apoe*^-^)^−/−^ mice, with NF-κB-dependent VCAM-1 expression and endothelial hyper-inflammation [125]. However, given the fact that Tie2 is also expressed by hematopoietic cells [126], the contributions of the senescent endothelium and senescent immune cells to atherogenesis need to be further specified. This concern has recently been addressed in a mouse model with TRF2 KO under the control of a more specific endothelial promoter (*Cdh5*-Cre). These mice with endothelial-restricted senescence exhibit enhanced endothelial production of ROS and inflammatory cytokines MCP-1 and IL-1β, impaired endothelial-dependent vasodilation, and perturbed glucose metabolism due to reduced microvascular perfusion and increased endothelial inflammation in metabolically active organs (white adipose tissue and liver) [127]. Senescent ECs exhibit impaired NO production, elevated adhesive molecule expression, and pro-inflammatory cytokine secretion. Consequently, inflammation-mediated endothelial dysfunction has been observed in aged rat carotid arteries and the adipose tissue of the resistant arteries of diet-induced obese mice, with dysfunction arising downstream of TNF-α elevation [128,129]. TNF-α can induce oxidative stress by activating NOX4 or inducing mitochondrial dysfunction, causing endothelial dysfunction, apoptosis, iNOS induction, leukocyte adhesion, and even senescence [130,131,132]. The inhibition of TNF-α improves flow-mediated arterial dilation in aged rats and reduces the expression of inflammatory markers ICAM-1 and iNOS [129,133].

Another inflammatory senescence-signaling cascade is the Notch pathway, which has been reported to inhibit EC growth, promote vascular hyperpermeability, and induce EC senescence in a context-specific manner [134,135]. Notch signaling is enhanced in atherosclerotic regions of aortas from mice and humans, and is activated in the ECs of older adults [134]. Notch activity is also increased in accelerated senescence in response to progerin expression [136]. Progerin is an alternatively spliced prelamin A protein that accelerates telomere shortening and exacerbates aging-related transcription [137]. The *LMNA* gene encodes prelamin A protein, which normally splices into mature lamin A, a major structural constituent of the nuclear envelope. A mutation of the *LMNA* gene that favors the accumulation of progerin’s spliced form is the genetic cause of Hutchison Gilford progeroid syndrome (HGPS), with early-onset endothelial dysfunction and extensive SMC loss in elastic arteries. Patients with HGPS frequently die from atherosclerosis and related complications in their first or second decades of life. Importantly, the expression of the progerin protein is also elevated in vascular tissues and cells from otherwise healthy aged humans [138]. Importantly, endothelial progerin accumulation not only causes downregulation of eNOS and endothelial dysfunction, but also induces EC oxidative stress, inflammation, and senescence [139,140].

In conclusion, targeting signaling pathways such as NF-κB and Notch, along with the specific neutralization of inflammatory cytokines like IL-1β or TNF-α, may provide novel therapeutic strategies to combat inflammation-mediated endothelial dysfunction and EC senescence in the context of atherosclerosis. Inflammation and EC senescence are summarized in Figure 2.

#### 2.2.3. Genomic Instability

Genomic instability, acquired via DNA damage as single- or double-strand DNA breaks or adducts over time, is an important mechanism that underlies the age-related accumulation of senescent cells in multiple organs, including elastic arteries [98,141]. Both external genotoxic stressors (oxidative stress and mechanical stress) [142,143] and telomere attrition are important triggers of DNA damage. Genomic instability in vascular ECs and SMCs has been recognized as a causal factor in senescence, vascular aging, and atherosclerosis [30,144,145]. Moreover, microsatellite instability and loss of heterozygosity, which is thought to be a consequence of non-productive repair of DNA breaks, has also been linked to atherosclerosis [146].

Telomere damage, including shortening or length-independent DNA lesions, is one of the best-established determinants of vascular cell senescence. Critically short telomeres trigger cellular senescence and limit EC proliferation, leading to impaired vascular repair and increased susceptibility to atherosclerosis [147,148]. Shelterin is a six-protein complex that protects telomere integrity at the end of chromosomes. Loss of the shelterin components with aging, inflammation, or oxidative stress causes telomere uncapping and excess DNA damage response, in turn eliciting prosenescent signals in ECs [8,149]. This mechanism, particularly via the loss of telomere-bound TRF2, a key shelterin protein, has been recently linked to in vivo EC shear response and senescence in a hyperlipidemic low-density lipoprotein receptor (*Ldlr*)^−/−^ model of mouse atherosclerosis [150].

Multiple non-telomeric DNA damage/repair defects have been demonstrated as triggering the signaling of EC senescence and aging. One of the best examples is nucleotide excision repair (NER). NER defects have been associated with endothelial dysfunction and the pathogenesis of atherosclerosis. The Excision Repair Cross-Complementation group 1 (ERCC1) and xeroderma pigmentosum group D (XPD), which play essential roles in NER, have been found to protect against EC senescence in mice, suggesting a causal relationship between defective NER and vascular aging [151]. Importantly, the single-nucleotide polymorphism (SNP) of DNA damage binding Protein 2 (DDB2), another NER component, has been linked to carotid–femoral pulse wave velocity, a hallmark of arterial aging, in over 20,000 participants of the AortaGen Consortium [151]. Nevertheless, a causal link between EC-specific DNA repair machinery defects and atherosclerosis has yet to be established. Genomic instability and EC senescence are summarized in Figure 3.

#### 2.2.4. Metabolite-Sensing Longevity Pathways

Metabolite-sensing pathways play a pivotal role in cellular energy homeostasis, nutrient sensing, and overall metabolic regulation, with strong roles in the development and progression of atherosclerosis. The three key players in these pathways are SIRT, AMPK, and mTOR. Notably, strong connections have been documented between these three metabolite-sensing signaling nodes, vascular cell senescence, and atherosclerosis [152,153,154].

##### Sirtuins

Sirtuins are a family of class III NAD^+^-dependent deacetylases with diverse roles in aging, inflammation, and stress responses [155]. Among the seven mammalian sirtuins (SIRT1–SIRT7), SIRT1 and SIRT6 have been widely studied in the context of EC senescence and atherosclerosis.

SIRT1 has been reported to play an endothelial-protective role, maintaining endothelial function, preventing endothelial senescence, and ameliorating atherosclerosis. The earliest experimental evidence has shown that chemical or siRNA inhibition of SIRT1 leads to senescence in HUVECs via enhancing the acetylation and activity of p53 [156]. Overexpression of SIRT1 prevents H_2_O_2_-induced EC senescence and downregulation of eNOS [156]. Subsequent studies have revealed that SIRT1 expression and activity decline with age in human vasculature. Such age-related SIRT1 repression is associated with human endothelial dysfunction, as determined by impaired flow-mediated vasodilation in the forearms [157]. Importantly, SIRT1 directly regulates NO expression and endothelial function in the arteries by deacetylating and subsequently activating eNOS [157]. Mice with endothelium-specific overexpression of SIRT1 are protected against diet-induced atherosclerosis in hyperlipidemic *Apoe^−/−^* mice, with improved aortic NO production independent of blood lipid and glucose levels [158]. Interestingly, global *Sirt1* heterozygous KO enhances endothelial inflammation, without affecting eNOS activity, in atherosclerotic *Apoe^−/−^* mice [159]. It has been proposed that the NF-κB subunit RelA/p65 is a direct target of the deacetylase of SIRT1, and its deacetylation suppresses the transcriptional activity of NF-κB on proinflammatory adhesive molecules ICAM-1 and VCAM-1 in ECs [159,160]. In addition to eNOS, p53, and NF-κB, SIRT1 can function through many other target proteins, including FOXOs [161] and hypoxia-inducible factor-1α (HIF1α) [162], via its deacetylase activity to improve cardiovascular health. A pharmacological activator of SIRT1, SRT1720, has been shown to extend mouse lifespan and healthspan [163] and to reverse endothelial inflammation, ROS production, and endothelial dysfunction in aged mice [164]. Multiple small-molecule SIRT1 activators have entered clinical trials, but their cardiovascular benefits in humans have not been clearly established [165,166]. However, hope remains based on strong evidence from human genetics. Several SNPs of the *SIRT1* gene have been associated with features of atherosclerotic CVDs, including carotid intima–media thickness and risk of coronary artery disease [167,168,169]. Notably, rs3758391 T→C, located in the *SIRT1* gene promoter, is associated with elevated *SIRT1* mRNA levels in patients with acute coronary syndromes [170].

SIRT6 is another well-documented sirtuin that can protect against EC senescence and atherosclerosis by promoting genomic stability, reducing inflammation, and epigenetically modulating age-related gene expression [171,172]. Global SIRT6 heterozygous KO (*Sirt6^+/−^*) exacerbates atherogenesis in *Apoe^−/−^* mice fed a high-fat diet, likely via derepressing the expression of proinflammatory adhesive molecules, such as VCAM-1, and promoting monocyte infiltration into the arterial wall [173]. Importantly, the endothelium-specific *Sirt6* KO impairs endothelium-dependent vasodilation and causes hypertension [173,174]. Multiple mouse models with endothelial *Sirt6* deficiency have generated critical evidence to support an anti-atherosclerotic role of SIRT6, although the in vivo senescence in these mice is not always clearly indicated [175,176]. At the cellular level, SIRT6 protects cultured human ECs from senescence via maintaining telomere integrity and reducing DNA damage [177], or by upregulating Forkhead box protein M1 (FOXM1) and promoting cell cycle progression [178].

In addition to SIRT1 and SIRT6, SIRT2 [179], SIRT3 [180], and SIRT7 [181,182] have also been reported to play anti-senescent roles in ECs. Taken together, emerging evidence suggests that sirtuin isoforms, particularly SIRT1 and SIRT6, play a protective role against EC senescence and atherosclerosis. They achieve this by modulating the oxidative stress response, inflammation, genomic stability, and epigenetics in vascular cells. Considering the complex clinical trial experience of SIRT1 activators, further research is needed to explore the potential therapeutic implications of these findings in the context of atherosclerosis and vascular aging.

##### AMPK

AMPK, a highly conserved heterotrimeric serine/threonine kinase, is an important energy-sensing signaling protein that integrates energy homeostasis, metabolism, and stress resistance [183]. In general, AMPK delays cellular senescence and ameliorates age-related signaling perturbations [184] in autophagy and mTOR [185,186] cascades. The signaling relationships between AMPK and autophagy or mTOR have been extensively studied and have proven to be very complex. AMPK activates autophagy via phosphorylating Unc-51-like kinase 1 (ULK1) or modulating autophagy-related transcription factors such as FOXOs. Ge et al. conducted an extensive review on the significant role of AMPK in the intricate connection between cellular autophagy and aging [184]. Moreover, AMPK can directly phosphorylate and inactivate mTOR, or do so indirectly via phosphorylating tuberous sclerosis complex 2 (TSC2) [184]. Interestingly, blunting age-related physiological dysfunction after mTOR inactivation also results in increased AMPK activity [187]. AMPK activation can also directly inhibit mTOR by phosphorylating Raptor, a crucial mTOR adaptor protein [188]. Additionally, AMPK signaling has been shown to be upregulated in ribosomal S6 protein kinase (*S6K1*) KO mice that have reduced mTOR signaling [187]. In ECs specifically, AMPK limits endothelial dysfunction by stimulating NO production. Mechanistically, AMPK enhances eNOS activity through Ras-related C3 botulinum toxin substrate 1 (Rac1)-AKT-dependent phosphorylation of eNOS [189]. Age is associated with declines in AMPK activity in mouse aortas and cerebral arteries [190]. Sustained activation of AMPK using AICAR, an AMP analog and AMPK agonist, improves the endothelium-dependent vasodilation of isolated carotid arteries of aged mice, and this effect is independent of NO bioavailability, suggesting a context-dependent vascular benefit of AMPK [190]. Furthermore, the amelioration of oxidative stress has been reproducibly reported as a convergent mechanism for the anti-aging and anti-senescence effects of AMPK in the endothelium [189,191,192,193]. On the other hand, it should be noted that hyperactivation of AMPK, through overexpression of liver kinase B1 (LKB1), is reported to induce senescence in cultured porcine aortic ECs [107]. In hyperlipidemic mouse models, activation of AMPK by AICAR or metformin has been reported to have an anti-atherogenic benefit, with the involvement of direct effects on the arterial wall, non-vascular organs, and immune cells [194,195]. For instance, metformin activation of AMPK improves mitochondrial biogenesis through interplay with SIRT1 and SIRT3 to antagonize replicative senescence of cultured ECs, and delays arterial senescence, inflammation, and spontaneous atherogenesis in *Apoe^−/−^* mice [196]. AMPK has a vaso-protective effect on endothelial aging and senescence via its pleiotropic signaling capacity involving autophagy, mTOR, sirtuins, eNOS, and ROS. However, in vivo evidence on the direct effects of AMPK on endothelial senescence in the context of atherosclerosis is still lacking.

##### mTOR

mTOR is a serine/threonine kinase which responds to nutrients or growth factors to modulate mRNA translation; protein synthesis; and cellular growth, proliferation, inflammation, and senescence [197]. Mounting studies have shown that decreasing mTOR signaling, either genetically or chemically with rapamycin or rapamycin-like drugs, can slow senescence and vascular aging and extend the lifespans of multiple organisms, including rodents [198,199,200]. Evidence has demonstrated that mTOR signaling is augmented in arteries from older mice which display endothelial dysfunction. Lifelong caloric restriction prevents augmented arterial mTOR signaling and endothelial dysfunction [201]. Activation of mTOR/S6K1 signaling has been demonstrated to play a causal role in endothelial senescence and endothelial dysfunction related to eNOS uncoupling in both a cell culture model and a rat aging model [202]. Furthermore, pilot studies suggested that dietary supplementation of rapamycin for 6 to 8 weeks improves NO and endothelial function in aged mice [203].

mTOR is a core component of two distinct multi-protein complexes: mTORC1, which contains mTOR, mammalian lethal with SEC13 protein 8 (mLST8), proline-rich AKT substrate of 40 kD (PRAS40), and DEP domain-containing mTOR-interacting protein (DEPTOR), and mTORC2, which contains mTOR, rapamycin-insensitive companion of mTOR (RICTOR), mammalian stress-activated protein kinase-interacting protein 1 (MSIN1), and protein observed with RICTOR 1 (PROTOR-1). These two complexes take part in endothelial senescence through different signaling cascades. mTORC1 relays the phosphoinositide 3-kinase (PI3K)-AKT signals evoked by replicative stress to NF-κB in cultured ECs to promote senescence-associated inflammatory secretion, such as IL-6, and paracrine senescence to impact proliferating neighbor cells [204]. Notably, such proinflammatory effects of mTOR on SASP have also been documented for tumor cells, involving diverse signaling cascades [205,206]. mTORC2 is also a prosenescent mediator in ECs when challenged with H_2_O_2_ [207]. Upon oxidative stress, mTORC2 acts upstream of AKT, which in turn suppresses CCAAT/enhancer binding protein alpha (CEBPα)-dependent expression of Nrf2 and its downstream antioxidant defense system [207]. siRNA knockdown of RICTOR diminishes the replicative and oxidative stress-induced senescence of HUVECs [207].

Attributable to the pleiotropic effects of mTOR on vascular cell activation and foam cell lipid metabolism, mTOR inhibition has been proposed as a promising therapeutic approach to atherosclerosis. Despite the established prosenescent and pro-aging roles of mTOR at the organ/tissue level, mTOR-mediated EC senescence has not been specifically investigated in the context of age-related vascular diseases. Recently, in a mouse model with tamoxifen-induced EC-specific KO of PRAS40, a negative regulatory component of mTORC1, endothelial inflammation and atherogenesis were exacerbated, and this was correlated with augmented mTORC1 signals [208]. Although the relationship between mTOR activity and endothelial senescence has not been specifically examined in these mouse atherosclerotic lesions, this study provides important evidence of mTOR modulation of endothelial function as a pathogenic determinant of atherosclerosis.

##### Klotho and FGF21

Klotho is a prominent anti-aging glycoprotein [209] serving as a coreceptor to facilitate the high-affinity binding of endocrine FGF19 family ligands (FGF19, FGF21, and FGF23) to their cognate FGF receptors. The main organ that Klotho directly targets is the kidney, where its protein expression is most abundant. As a comparison, conflicting data show that the vascular expression of Klotho mRNA and protein ranges from non-detectable to detectable levels. Nevertheless, Klotho has a wide spectrum of cardiovascular benefits, and genetic deficiency of Klotho in mice causes arterial medial calcification, intimal hyperplasia, hypertension, endothelial dysfunction, and even reduced lifespan [210]. Notably, low serum Klotho levels are associated with subclinical atherosclerosis or carotid atherosclerotic plaque burden in patients with type 1 diabetes or chronic kidney disease [211,212,213,214]. Furthermore, the serum and aortic levels of Klotho negatively correlate with coronary artery disease severity, independent of other established CVD risk factors [215]. At the cellular level, Klotho is reported to prevent Ang II- or IL-1β-induced senescence of HUVECs via upregulating the cytoprotective Nrf2-heme oxygenase-1 pathway [216]. In addition, Klotho can inhibit the transcriptional activity of NF-κB on cytokine genes, including IL-6 and TNF-α, in HUVECs [217,218]. Recently, Miao et al. have demonstrated that Klotho suppresses Ang II-induced endothelial senescence via inhibiting the wingless-type MMTV integration site family member 3A (Wnt3a)-glycogen synthase kinase-3β (GSK3β)-mTOR pathway, and established a strong association between low serum Klotho levels and an increased risk of major adverse cardiovascular events in 295 patients who underwent a percutaneous coronary intervention with 12 months of follow-up [219].

The anti-aging benefits of Klotho largely depend on endocrine FGFs, among which FGF21 is reported to play an anti-senescent role in ECs. FGF21 is a starvation hormone that induces stress responses by activating the sympathetic nervous system and the hypothalamus–pituitary–adrenal axis [220]. Importantly, growing evidence indicates that exogenous FGF21 has anti-atherosclerotic effects via its actions on lipid metabolism, oxidative stress, inflammation, and cell vitality (senescence and death) [221]. Importantly, FGF21 deficiency exacerbates the development of atherosclerotic plaque formation in *Apoe^−/−^* mice. This athero-protective effect of FGF21 is proposed to act via the inhibition of sterol regulatory element-binding protein-2 (SREBP2) and cholesterol biosynthesis in mouse liver [222]. FGF21 administration reverses endothelial dysfunction under oxidative stress in *Apoe^−/−^* mice and improves endothelial progenitor cell functions via the AKT/eNOS/NO pathway [223]. Similarly, FGF21 can improve endothelial dysfunction in type 1 diabetes, and the molecular mechanism involves AMPK activation [224]. Furthermore, FGF21 alleviates EC death in the form of apoptosis or pyroptosis, constituting another cytoprotective mechanism to curb atherogenesis [225,226,227]. Despite the accumulating evidence on the endothelial benefits of FGF21, the direct relationship between FGF21 and endothelial senescence remains poorly defined. Yan et al. reported that FGF21 protects HUVECs from H_2_O_2_-induced DNA damage, inflammation, and senescence in a SIRT1-dependent manner [228]. In addition, FGF21 can normalize the high-glucose-compromised vitality, migration, and eNOS activity of HUVECs, also lending support to an anti-senescent role of FGF21 [229]. Taken together, these data support the notion that Klotho and FGF21 have powerful vaso-protective effects during aging and under metabolic stress, and some of these benefits may act through the amelioration of endothelial senescence. Metabolite-sensing longevity pathways and EC senescence are summarized in Figure 4.

#### 2.2.5. Activated Renin-Angiotensin System (RAS)

RAS is an essential regulatory component of blood pressure and vascular tone. Chronic activation of RAS contributes to vascular aging via directly modifying vascular cell functions, including enhancing endothelial senescence. Ang II acts through the angiotensin type 1 and type 2 receptors (AT1R and AT2R). Age is associated with elevated expression of Ang II, angiotensin-converting enzyme 1 (ACE1), and AT1R in multiple tissues and cell types, including arterial EC and SMCs [230]. AT1R activation on ECs and SMCs results in NAPDH oxidase-mediated ROS production, inflammation, fibrotic matrix remodeling, and vasoconstriction, whereas the activation of AT2R inhibits cell proliferation, inflammation, and fibrosis [9]. Moreover, Ang II can trigger mitochondrial dysfunction and enhance the production of mitochondrial ROS (mitoROS) [231], which is likely related to the mitochondrial adaptor protein p66Shc [232]. Both cytoplasmic and mitochondrial ROS decrease endothelial NO formation. The reduced NO bioavailability, in synergy with Ang II/AT1R/Rho-associated protein kinase (ROCK)-induced SMC contraction, compromises the endothelium-dependent vasodilation of arteries [233]. Importantly, global KO of *AT1A*, the closest murine homolog to human AT1R, prolongs mouse lifespan; limits oxidative damage in the heart, kidneys, and aorta, and diminishes age-related aortic damage, such as elastin fragmentation and atherosclerotic foam cell formation [234]. A meta-analysis of 2366 cases and 2414 controls demonstrated that the AT1R A1166C (rs5186) polymorphism is associated with a risk of coronary heart disease in the East Asian population, with an odds ratio of 1.50 for C versus A [235]. This SNP has also been linked with essential hypertension [236,237,238,239]. Pharmacological inhibition of RAS, especially AT1R and ACE1, which serve as rate-limiting steps of Ang II production, has been routinely prescribed as an anti-hypertensive therapy and proposed as a potential approach by which to delay vascular aging [240,241]. In this regard, mounting evidence shows that losartan, an AT1R blocker, reduces arterial stiffness, as measured by pulse wave velocity or ex vivo myography, in hypertensive patients [242,243,244]. In particular, a recent clinical study including 308 elderly, hypertensive subjects and with a 10-year follow-up demonstrated that long-term use of losartan is associated with a reduced risk of acute coronary syndrome [245]. More importantly, such cardiovascular benefits of RAS inhibition can be generalized to normotensive atherosclerotic patients, as established by a meta-analysis of 13 trials and 80,594 patients [245]. Animal experiments also provide striking data to support the life-extending effects of RAS inhibition. Treatment with ACE inhibitors or losartan significantly increases the lifespan of male Wistar rats by ~20% [246], partially via sustaining endothelial NO production during aging [247,248].

ROS, derived from NADPH oxidase or mitochondria, are recognized as major prosenescent signal mediators by Ang II. Furthermore, ROS-activated NF-κB is proposed to mediate surface presentation of adhesive molecules and secretion of proinflammatory cytokines by senescent ECs [249,250]. Beyond ROS and RNS, in recent years, more molecular targets have been identified to regulate Ang II-induced endothelial senescence: sodium-glucose cotransporter (SGLT) 1 and 2 are among the most intriguing, given the recent breakthrough of using a SGLT2 inhibitor as treatment for type 2 diabetes and its promising cardiovascular benefits [251,252,253]. Ang II can elevate the expression of SGLT1 and SGLT2 in cultured porcine coronary arterial ECs and rat aortic arch [254]. SGLT inhibitors, sotagliflozin or empagliflozin, almost completely block Ang II-induced EC senescence in vitro and normalize Ang II-induced endothelial inflammation and eNOS inactivation [254]. Interestingly, empagliflozin can also restore endothelial NO production and antagonize endothelial senescence induced by H_2_O_2_ or high glucose, with a potential suppressive effect on the protein levels of ACE1 and AT1R [255], suggesting a critical RAS-SGLT interplay in endothelial senescence. In summary, RAS, especially Ang II, is an important determinant of endothelial aging. Through provoking endothelial ROS, inflammation, death, and senescence, Ang II-mediated endothelial dysfunction confers a substantial risk of atherogenesis. Activated RAS system and EC senescence are summarized in Figure 5.

#### 2.2.6. Uric Acid

An elevated uric acid level in serum is recognized as a risk factor for endothelial dysfunction and CVD [256,257]. As discussed in previous sections, uric acid is an enzymatic product of XO, and ROS are considered as integral components of hyperuricemia-associated vascular stress. The concomitant generation of superoxide and uric acid reacts with NO directly to decrease the bioavailability of NO and the production of peroxynitrite. High uric acid levels in serum are observed in aged individuals [258], and are associated with compromised endothelial function in patients with obesity, chronic kidney disease, and hypertension [259,260,261,262,263]. Remarkably, elevated uric acid levels in serum were found to be associated with CVD in a cohort of 675 patients, and this association was attributed to reduced vascular NO production [264]. A causal link between aortic XO activity and endothelial dysfunction or arterial stiffening has been established in mice with diet-induced cardiometabolic stress [265,266]. Importantly, uric acid-induced oxidative stress can activate RAS, in turn mediating endothelial dysfunction and EC senescence [267]. Uric acid also blocks the promitogenic effects of epidermal growth factor (EGF), thereby contributing to EC senescence [268]. It should be noted that the prosenescent effects of uric acid have only been observed in cultured HUVECs so far. It remains to be determined whether uric acid impacts endothelial senescence in atherosclerotic lesions. On the other hand, uric acid has also been reported to function as a scavenger of peroxynitrite to protect endothelium-dependent vasodilation. For example, the preincubation of mesenteric microvessels from healthy elderly subjects (>60 years of age) with uric acid partially restored endothelium-dependent vasodilation, to a similar extent as SOD or TEMPOL (an SOD mimetic) [269]. Uric acid is an important stress signal for the endothelium with vascular aging and diseases. However, whether uric acid functionally modifies endothelial senescence in the context of atherosclerosis warrants further study. Uric acid and EC senescence are summarized in Figure 6.

#### 2.2.7. Polyploidization

Polyploidy is a cell state with more than two sets of chromosomes as a consequence of DNA replication without cell division, such as endoreduplication, cytokinesis failure, or cell fusion [270]. Polyploidization of vascular cells has been associated with a spectrum of pathological conditions, such as hypertension, stroke, atherosclerosis, and vascular aging [271,272,273,274,275]. Importantly, polyploidization of both SMCs and ECs is recognized as a precursor to cellular senescence [272,276,277]. Polyploid SMCs and ECs accumulate in aged arteries [271,272,273]. Moreover, electron microscopic studies have shown that the proportion of polyploid ECs correlates with the severity of atherosclerosis in the human aorta and iliac artery, as reported by Dr. Smirnov’s group [271] and Dr. Tokunaga’s group [273]. In particular, Dr. Borradaile and Dr. Pickering performed a series of careful studies on the relationship between polyploidy and senescence in human arterial ECs [276,277]. Polyploid ECs have defective cell cycle profiles featuring both G1 and mitotic arrest [277]. Between the cumulative population doubling 15 and 40, the prevalence of EC polyploidy is higher than that of senescence [276,277]. Although the exact cell cycle mechanism for polyploid ECs entering senescent status remains elusive, several critical aspects of endothelial dysfunction have been associated with polyploidization. Polyploid ECs have elevated ROS levels and reduced eNOS activity, correlated with increased expression of inflammatory adhesive protein ICAM-1. Moreover, transcriptomic analysis reveals a perturbed ECM profile and enhanced inflammation. Intriguingly, the overexpression of nicotinamide phosphoribosyltransferase (Nampt) and activation of SIRT1 rescues polyploidy-associated endothelial dysfunction, which is reminiscent of the anti-senescent role of this cascade [277]. In brief, age-driven EC polyploidization precedes senescence, and the polyploid phenotypes including cell cycle arrest, elevated ROS, and inflammation, very likely contribute to senescence and adverse vascular remodeling. Does this polyploidy–senescence link hold true in vivo? What are the relative contributions to vascular aging and atherosclerosis? These important questions remain unanswered and necessitate the development of an animal model that enables the genetic manipulation of endothelial cell ploidy, as in a hepatocyte study [278]. Polyploidization and EC senescence are summarized in Figure 7.

### 2.3. Senolysis—Therapeutic Targeting of Senescent ECs

#### 2.3.1. Pharmacological Senolytic Approaches

Normally, senescent cells can be cleared by immune cells, such as natural killer cells and T lymphocytes [279,280,281]. However, with age or under atherosclerotic conditions, senescent ECs, SMCs, fibroblasts, and immune cells accumulate in the arterial walls [34,37,282,283]. The increase in senescence burden not only impairs the replicative and reparative capacity of SMCs and ECs, but also adversely impacts non-senescent neighbors and even distant organs via SASP [284,285]. Persistent cellular senescence and SASP are recognized as driving forces for chronic inflammation and adverse arterial remodeling in atherosclerosis. Therefore, senolysis, the selective elimination of senescent cells, is considered to be a novel therapeutic approach in addition to canonical lipid-lowering therapy and recently developed anti-inflammatory therapy.

The conceptual framework of senolysis was first proposed by Dr. Kirkland’s group in 2015 [286]. Through in-depth transcriptomic analysis of senescent and non-senescent cells, an upregulated anti-apoptotic gene set was identified as a common transcript signature of senescent preadipocytes and HUVECs via radiation-induced genomic instability or replicative aging. A targeted drug screening revealed that a combination of dasatinib and quercetin preferentially reduces the viability of cultured senescent preadipocytes and HUVECs, as compared with their non-senescent counterparts. This dasatinb/quercetin combination is able to reduce the senescent cell burden in white adipose tissue and liver of aged, 24-month-old mice, improve the physical performance of mouse limb muscles injured by radiation exposure, and, amazingly, demonstrate notable cardiovascular benefits, including improved left ventricular ejection fraction and endothelium-dependent vasorelaxation [286]. Since, this first-generation senolytic drug has shown considerable benefits in animal models of a wide spectrum of age-related diseases, such as cardiometabolic syndromes, including diabetes [287], obesity [288] and atherosclerosis [289]; cardiac dysfunction [290]; neurological deficits [291,292]; sarcopenia [293]; joint and skeletal disorders [294,295]; and even COVID-19 [296,297]. Importantly, this senolytic regimen has been evaluated in multiple phase 1 and 2 trials (NCT 02874989, 04785300, 02848131, 02652052, and 04685590).

In the past several years, the following studies have greatly expanded the realm of senolytic reagents. An increasing number of chemicals targeting anti-apoptotic BCL-2 signaling cascade have been identified or repurposed as senolytic drugs, including navitoclax (also known as ABT-263), Fisetin, UBX0101 (also known as nutlin-3a), A1331852, and UBX1967 [298]. Among these senolytics, navitoclax has demonstrated extensive cardiovascular benefits in animal models of chemotherapy-induced cardiomyopathy [299], angiotensin II-induced heart failure [300], and myocardial infarction [301,302]. Importantly, navitoclax has been carefully tested in mouse models of atherosclerosis [37,303,304]. In 2016, Dr. van Deursen’s group demonstrated that intraperitoneal injection with 100 mg/kg navitoclax drastically reduced Sudan IV+ early atherosclerotic lesions in the entire aorta by 70% in *Ldlr*^−/−^ mice fed a high-fat diet [37]. In 2021, Dr. Martin Bennett’s group employed single-cell RNA sequencing to demonstrate that navitoclax (oral gavage, 50 mg/kg) could significantly reduce atherogenesis at the aortic root and in the descending thoracic aorta in *Apoe*^−/−^ mice fed a high-fat diet [303]. The anti-atherosclerotic benefit of navitoclax was argued to act mainly through selective killing of dedifferentiated SMCs in the lesions. Furthermore, a very recent preprint from Dr. Gary Owens’ group has raised important concerns regarding the plaque-destabilizing effect of navitoclax [304]. Intraperitoneal injection with 100 mg/kg navitoclax (the same dose as the 2016 study by Dr. van Deursen’s group) in *Apoe*^−/−^ mice fed a Western diet caused >50% mortality by 40 days after the onset of senolytic treatment. This is likely due to reduced plaque stability in the brachiocephalic artery. Single-cell RNA sequencing and SMC and EC lineage tracing revealed a 90% reduction in SMC content and increased endothelial contributions to the lesion via endothelial-to-mesenchymal transition. Such a dramatic alteration in the cellular origin of atherosclerotic plaques likely underlies the observed lethal plaque instability. Different drug sources and delivery methods, or different mouse lines and diets, could all contribute to the discrepancy among these navitoclax and atherosclerosis studies. Nevertheless, the Owens group’s study indicates the necessity of careful testing of senolytics across different mouse models and by multiple independent groups.

Other molecules have also been explored as senolytic targets, such as heat shock protein 90 (HSP90) [305]. Mechanistically, HSP90 protects AKT from proteasomal degradation to maintain survival signals in senescent cells. Inhibitors of HSP90, such as 17-DMAG, a geldanamycin derivative, lead to the selective induction of apoptosis in senescent cells, including HUVECs [305]. However, this has not been tested in animals with vascular aging or diseases yet. Another intriguing class of senolytic drugs is cardiac glycosides, a class of medicines to treat congestive heart failure and atrial fibrillation, that targets the Na+/K+ ATPase pump on the cell surface [306,307]. Although its efficacy on senescent ECs is yet to be determined, the clinical safety profiles of digoxin and ouabain make cardiac glycosides a tantalizing novel approach to reduce the senescent burden in diseased vessel walls.

Senescent cells have increased lysosomal biomass and β-galactosidase activity. One senolytic approach takes advantage of this feature to encapsulate cytotoxic drugs with galacto-oligosaccharides, or to covalently modify drugs with a galactose moiety [308,309,310]. This selectively enhances drug delivery and cytotoxicity to senescent cells. Notably, a preferential targeting of senescent lung cells, including ECs, in healthy and fibrotic mouse lungs by galacto-oligosaccharide nanoparticles has been demonstrated [308]. It will be intriguing to study the functional performance of this drug delivery system in an animal model of atherosclerosis.

Recent studies have demonstrated that lysosome dysfunction in senescent cells is associated with enhanced lysosomal permeability and cytosolic acidification. As a countermeasure, senescent cells upregulate GLS1 expression and glutaminolysis, which leads to increased ammonia production. This process can neutralize pH value lowering to improve senescent cell survival. Notably, the pharmacological inhibition of glutaminolysis by BPTES accelerates the replicative senescence of cultured ECs. However, the functional role of EC glutaminolysis is uncharted to date. Meaningful knowledge might be borrowed from vascular remodeling in pulmonary hypertension, another disease featuring EC senescence [311,312]. Bertero et al. reported that ECM stiffening increases GLS1 expression and enhances glutaminolytic flux and proliferation in cultured pulmonary arterial ECs [28]. Pharmacological inhibitors of GLS1, i.e., C968 and CB-839, reduces the proliferation of pulmonary arteriolar ECs and SMCs, thereby reducing pulmonary-hypertension-associated right ventricular dysfunction [28]. Similar matrix stiffening has been observed in hypertensive and atherosclerotic arterial walls, but no systematic study on EC glutaminolysis has been published to date. Therefore, the role of endothelial glutaminolysis in vascular aging and atherosclerosis remains a tantalizing candidate target for senolysis.

Senolysis is a fast-evolving field of research and innovation. The list of senolytic strategies has been growing to include the development of engineered chimeric antigen receptor/CAR T cells [313] that recognize uPAR-expressing senescent cells, as well as senolytic vaccination [314,315] that targets glycoprotein nonmetastatic melanoma protein B or CD153. Further studies are warranted to examine and optimize the efficacy of these novel senolytic methods in the vascular system.

#### 2.3.2. Genetical Senolytic Approaches

Many high-quality studies have performed comparative studies between pharmacological agents and non-pharmacological, genetic approaches that enable the specific ablation of senescent cells in mouse organs. INK-ATTAC is the earliest mouse model, created by Dr. van Deursen’s group, that allows for the visualization and targeted ablation of senescent cells [316]. The INK-ATTAC transgene contains a p16^Ink4a^ promoter and an expression cassette for FK509 binding protein (FKBP)-caspase-8 fusion protein, followed by an internal ribosome entry site (IRES)-segregated sequence encoding eGFP protein. In this mouse model, senescent cells, as marked by high p16 expression, express the FKBP-caspase-8 protein. These proteins remain inactive until the addition of AP20187, a synthetic drug that induces the dimerization and activation of FKBP-caspase-8, leading to apoptosis of senescent cells. This mouse model was first employed to study the dynamics and function of senescent cells in BubR1 progeroid mice. p16-3MR is another similar senolytic mouse model, created by Dr. Judi Campisi’s group [122]. This mouse model was first applied in a study on senescent ECs and mesenchymal cells in skin wound healing. Both mouse models have been widely employed to investigate the causal roles of cellular senescence in cardiovascular diseases, including vascular aging and atherosclerosis [37,289,303]. For example, Roos et al. demonstrated both a dasatinib/quercetin cocktail and the ablation of p16-high senescent cells using INK-ATTAC mice show a comparable benefit in reverting the vasomotor functions of aged mice [289]. In another seminal study, p16-3MR mice crossbred with *Ldlr*^−/−^ mice were used to eliminate >90% of the senescent ECs, SMCs, and macrophages in atherosclerotic lesions [37]. This efficient senolytic approach led to a reduction of around 50% in the atherosclerotic burden induced by a high-fat diet, supporting a detrimental role of senescent cells in atherogenesis.

Further, more senolytic mouse models have been recently used, including a particularly notable p21-driven senolytic mouse model which was created and used to study the p21-high senescent cell population in a mouse model of obesity, including microvascular ECs [317,318]. As p16 and p21 may represent distinct senescent cell populations [319], a comparative study between p16- and p21-high senescent cell ablation in the context of atherosclerosis would be highly intriguing.

#### 2.3.3. Challenges of Senolysis

Despite the ongoing promising progress in senolytic research, there are still many considerations from bench to bed. First, most senolytics are systemically administrated, and the potential side effects of senolytics in different vascular beds and organ systems need to be carefully examined. For example, navitoclax is well-known to cause thrombocytopenia, a condition characterized by a decrease in platelet count. Based on the registry of ClinicalTrials.gov, there are over 20 clinical studies regarding senolytic therapy, both in progress and completed. There are even more trials of drugs that possess senolytic property, although they do not aim to eliminate senescent cells. Knowledge from these trials will be critical to understanding and managing any adverse effects of senolytics. Notably, there are no registered trials of senolytics for atherosclerotic cardiovascular diseases yet. Second, the cell-type-specific effects of senolytic treatment remain to be defined. Atherosclerosis has a complex pathogenic course involving multiple cell types. Although senescent ECs are generally considered pro-atherogenic, the in vivo outcomes of senescent EC elimination in the context of atherosclerosis are far from clear. In this regard, the genetic ablation of p16-high senescent liver sinusoidal ECs in aged mice raises critical concerns for senolytics [320]. Recently, preclinical studies of atherosclerosis have begun to integrate senolytic treatment with single-cell omics. As mentioned earlier, Garrido et al. showed that navitoclax reduces the atherosclerotic lesion burden and necrotic core size in hyperlipidemic *Apoe*^−/−^ mice [303]. This benefit is argued to act through the selective killing of SMCs, as the authors detected few ECs that strongly expressed p16 transcripts in single-cell RNA sequencing of mouse atherosclerotic plaques. However, this study alone is not sufficient to exclude the therapeutic value of targeting senescent ECs in atherosclerosis, as the p16 transcript level can be difficult to detect in single-cell RNA sequencing assays and may not acutely represent p16 protein abundance [321,322]. More experiments are needed to directly compare the effects of selective ablation of ECs versus other cell types in the same atherosclerotic animals. Understanding such cell-type-specific impacts of senolytics will be essential to advancing this promising regimen into the management of atherosclerotic cardiovascular diseases safely and efficiently.

## 3. Vascular EC Death in Atherosclerosis

### 3.1. EC Death and Atherosclerosis

The roles of various forms of regulated EC death, including apoptosis, autophagic cell death, NETosis-related cell death, ferroptosis, pyroptosis, and necroptosis, in the development of atherosclerosis have been investigated extensively, as summarized in Figure 8 and Table 2. Various proatherogenic signals, such as low-density lipoprotein (LDL), hyperglycemia, oxidative stress, and low shear stress, have been shown to induce EC death. Conversely, several athero-protective factors, like estrogen and NO, have been demonstrated to prevent EC death. The long-term influence of death-inducing factors causes a substantial loss of EC, leading to not only systemic endothelial dysfunction, but also plaque instability or surface erosion in atherosclerotic lesions locally. However, each mode of cell death is associated with the specific cellular signaling cascades and unique impacts on atherosclerosis, constituting a plethora of potential therapeutic targets. Before focusing on the death of ECs, we must emphasize the fact that multiple cell types have been reported to undergo cell death in atherosclerotic lesions, and the deaths of different cell types may yield various outcomes. For example, apoptosis of macrophages and foam cells precipitates cholesterol crystals, which enlarge the necrotic core and destabilize plaques [323]. Apoptosis of SMCs is frequently associated with the weakening of collagen fibrils, thinning of the fibrous cap, and calcification of lesions [324]. Non-apoptotic death has also been extensively characterized for macrophages and SMCs in atherosclerosis [325]. Herein, we will specifically discuss several well-studied forms of EC death in the context of atherosclerosis.

### 3.2. Autophagy—Protective and Detrimental

Autophagy is an essential cytoprotective process that degrades intracellular organelles and recycles the protein and lipids [340]. Genetic activation of autophagy by overexpression of ATG5, a key regulator of autophagosome formation, can even extend the lifespan of mice, partially through increasing cellular resistance to oxidative stress [341]. Notably, autophagic flux is critical to maintaining vascular cell homeostasis, and the decline in EC autophagy contributes to arterial aging [342]. Autophagy-enhancing agents, such as trehalose, spermidine, and rapamycin, revert the phenotypes of aged arteries in mice, including reduced vascular NO levels, elevated endothelial inflammation, and impaired endothelium-dependent vasodilation [199,342,343].

At homeostasis or in response to mild-to-moderate vascular stress, EC autophagy is generally considered to be vasoprotective [344,345]. This is attributed to its ability to balance the status of cellular redox and bioenergetics, which is central to proper EC functions and survival [346]. Multiple triggering signals for autophagy are present during aging, such as ROS, oxLDL, advanced glycation end-products (AGEs), and shear stress [347,348,349,350,351]. Specifically, autophagy can protect EC metabolism, NO production, and survival to restrict chronic vascular inflammation and atherogenesis, with control of the level of cellular ROS as a main mechanism [351,352,353]. In an inflammatory environment, intense autophagy in ECs has been found to reduce the expression of endothelial adhesion molecules, such as CD31 and VE-cadherin [354]. This process limits the migration of neutrophils across the endothelium, ultimately disrupting the cycle of inflammation. Moreover, EC autophagy mediates the vasoprotective effects of laminar shear stress, suppresses EC inflammation, apoptosis, and senescence, and alleviates the atherosclerotic burden in hyperlipidemic mice [355].

At the molecular level, SIRT1 has been recognized as a main signaling mediator of protective autophagy in ECs. Physiological, arterial-type laminar flow induces ROS in ECs, which stimulates EC autophagy through flow-activated SIRT1 and TFEB, a master transcriptional regulator of lysosomal biogenesis [356]. SIRT1 takes part in different stages of autophagic flux, ranging from autophagosome initiation to lysosomal degradation, which has been extensively reviewed elsewhere [357,358]. Herein, we briefly focus on the steps that have been experimentally validated in ECs. In HUVECs, SIRT1-dependent AMPK activation, likely via deacetylating/activating LKB1, inhibits mTORC1, in turn stimulating autophagy [359]. As mTOR is a master negative regulator of all main stages of autophagy [360,361], further studies are warranted to dissect the specific signaling role of the SIRT1-mTOR axis in EC autophagy. In addition to regulating the LKB1-AMPK-mTOR pathway, the deacetylase activity of SIRT1 works on a diverse range of pro-autophagic proteins. For example, upon nutrient starvation, SIRT1 deacetylates ATG5, ATG7, and ATG12 to facilitate the elongation of autophagic vesicles [362]. Importantly, the SIRT1-ATG5 link has been studied in oxLDL- and shear-stress-induced EC autophagy [363,364]. Another well-established pro-autophagic role of SIRT1 is that SIRT1-deactylated FOXO1 transcriptionally activates Rab7, which mediates autophagosome–lysosome fusion. The importance of the SIRT1-FOXO1 cascade in protective autophagy in ECs has been experimentally validated by several independent groups [365,366]. Interestingly, SIRT1 itself is susceptible to autophagic degradation, a process that contributes to the loss of SIRT1 during senescence and aging [367]. However, whether this bidirectional control between SIRT1 and autophagy functions in the aged endothelium is unknown.

On the other hand, autophagy also has broad adverse impacts on atherosclerosis via its profound detrimental effects on ECs, macrophages, and arterial SMCs, as it affects pathogenesis from the early to the advanced stages [368]. Critically, overactivation of autophagy can induce cell death, including death of EC, in a form distinct from other modes of death like apoptosis, necroptosis, and necrosis [369,370]. In 2003, Chau et al. demonstrated that endostatin, a collagen VIII cleaved fragment, can induce autophagic EC death in a ROS- and caspase-independent manner [371]. Importantly, serum endostatin levels correlate with subclinical atherosclerosis, as assessed by carotid intima–media thickness in a cohort of 648 healthy Japanese subjects [372], cardiovascular mortality in two Swedish community-based cohorts of 1689 elderly subjects [373], and mortality and severe disability after ischemic stroke in 3463 Chinese subjects [374]. However, the in vivo roles of endostatin in autophagic EC death remain undefined, although endostatin is reported to slow down mouse atherogenesis due to its anti-angiogenic effect or its capacity to interfere with LDL retention in intima [375,376]. Subsequent studies have identified multiple stress signals that trigger autophagic cell death, including ROS, cigarette smoke extract, glucose deprivation, and bisphosphonates [377,378,379,380]. Despite these advancements, in vivo evidence on autophagic EC death, especially its relevance to atherosclerosis, remains to be firmly investigated and established.

### 3.3. Apoptosis

Apoptosis is one of the most widely studied forms of programmed cell death, and it has diverse, context-dependent roles during development, homeostasis, and disease [381]. Apoptosis can be initiated via two main pathways: the extrinsic and intrinsic pathways. The extrinsic pathway is triggered by extracellular stress signals. Notably, many pro-atherogenic signals are also pro-apoptotic for ECs, including oxLDL, Ang II, TNF-α, homocysteine, and disturbed blood flow [382]. External stress triggers the assembly of a death-inducing signaling complex that activates caspase-8, which in turn cleaves and activates caspase-3, a major executor of apoptosis featuring chromatin condensation and DNA fragmentation [383]. In vivo, the remnants of apoptotic cells can be quickly cleared by phagocytic cells [381]. The intrinsic pathway is activated by perturbed intracellular signals, such as genomic instability due to DNA damage or mitotic defects, oxidative stress, or mitochondrial damage. This pathway mainly involves the release of pro-apoptotic proteins, typically cytochrome c, from the mitochondria into the cytoplasm. This then promotes the assembly of an apoptosome comprising APAF-1 and procaspase-9 for the downstream activation of caspase-3 and apoptotic cell death [384]. It needs to be noted that, despite being initially proposed as distinctive, the extrinsic and intrinsic apoptosis pathways have significant crosstalk and overlap. For instance, TNF-α not only activates tumor necrosis factor receptor 1 (TNFR1) to recruit tumor necrosis factor receptor type 1-associated DEATH domain (TRADD) for the assembly of the death-inducing signaling complex, but also causes mitochondrial outer membrane permeabilization (MOMP) and the release of cytochrome c [385].

Near half a century ago, increased apoptosis of ECs was found at the pig aortic arch [386], the athero-prone aortic zone. Importantly, accumulating evidence has revealed that EC apoptosis is an early event in the development of atherosclerosis. Tricot et al. observed increased EC apoptosis in 42 human carotid atherosclerotic plaques, with a preferential localization of apoptotic EC in the downstream parts of the lesions, where the blood flow becomes slow and complex [387]. Redox status is one of the main determinants of EC apoptosis during atherogenesis [388], with excess ROS derived from NADPH oxidase and dysfunctional mitochondria as a major driver of EC apoptosis [389,390]. Moreover, pro-atherogenic stressors, such as oxLDL, Ang II, hyperglycemia, pro-inflammatory cytokine TNFα, and low shear stress, are shown to induce EC apoptosis [381]. In contrast, athero-protective factors, such as estrogen, NO, and arterial-type laminar shear stress, prevent EC apoptosis [391,392].

OxLDL is a key biomarker for atherosclerosis, and it induces chronic endothelial inflammation, endothelial dysfunction, and apoptosis [393]. In 1997, Dimmeler et al. identified that the oxLDL-induced death process in HUVECs is a mode of caspase-3-dependent apoptosis [394]. Subsequently, diverse molecular cascades have been identified for oxLDL-mediated apoptosis. Harada-Shiba et al. reported that oxLDL induces a rapid accumulation of ceramide and superoxide to promote the apoptosis of HUVECs [395]. This pro-apoptotic effect is mediated by the oxysterol, but not phospholipid, fraction of oxLDL [395]. At the same time, Sata et al. showed that oxLDL sensitizes Fas-mediated EC apoptosis [396]. Ca^2+^ activity is also an important intracellular mediator for cytotoxic signals of oxLDL [397]. OxLDL-induced Ca^2+^ mobilization and subsequent cell death are inhibited by athero-protective high-density lipoprotein (HDL) or delipidated apolipoprotein A (apoA) [397]. Notably, lipoxygenase-1 (LOX-1), a lectin-like endothelial receptor for oxLDL, is transcriptionally upregulated by oxLDL itself, and causes EC apoptosis in an NF-κB-dependent manner [398]. Subsequent studies greatly expanded the spectrum of oxLDL-elicited apoptotic cascades, ranging from surface receptors, stress-sensing kinases, and transcription factors to non-coding RNAs. One intriguing transcription factor is Krüppel-like Factor 2 (KLF2), a well-documented mechanotransducer for shear stress in ECs [399]. KLF2, whose expression is elevated by laminar flow [400], executes its athero-protective effects via transcriptional activation of eNOS and thrombomodulin, or repression of inflammatory mediators including E-selectin, MCP-1, VCAM-1, and plasminogen activator inhibitor-1 (PAI1) [399]. Notably, an anti-apoptotic role of KLF2 has been reported for ECs. Zhang et al. reported that protein tyrosine phosphatase 1B (PTB1B) knockdown can prevent oxLDL-induced inflammatory injury and dysfunction in ECs, which is regulated at least in part by the AMPK/SIRT1 signaling pathway through KLF2 [401]. Expression of the E3 ubiquitin ligase 3-hydroxy-3-methylglutaryl reductase degradation (HRD1) was significantly decreased in atherosclerotic intima. Mechanistically, decreases in HRD1 levels in ECs can be caused by oxLDL. Elevated expression of HRD1 inhibits the EC apoptosis induced by oxLDL, potentially via promoting the ubiquitination–proteasomal degradation of LOX-1, the oxLDL receptor [402]. Importantly, KLF2 has been identified as a transcriptional activator of HRD1 [402]. These studies support the prevention of EC apoptosis as a potential athero-protective mechanism of KLF2.

While laminar shear stress stabilizes the endothelial barrier, sustains eNOS activation, and promotes endothelial quiescence and survival, disturbed flow, an atherogenic risk factor, evokes endothelial inflammation, permeability, and apoptosis [403,404,405,406]. Several mechanisms have been discovered for disturbed flow-induced endothelial apoptosis. Disturbed flow leads to sustained X-box binding protein 1 (XBP1) activation and splicing, one branch of ER stress responses, thereby inducing apoptosis of ECs [406]. Adenovirus-mediated XBP1 overexpression results in the development of in situ EC apoptosis and atherosclerotic lesions in an *Apoe^−/−^* mouse model of aortic graft [407]. Moreover, disturbed flow increases endothelial peroxynitrite production, which in turns activates a cascade of the PKCζ/PIASy/p53 SUMOylation/BCL-2 pathway to execute apoptosis in vitro and in vivo [408]. Compared with the athero-protective greater curvature of the mouse aortic arch, the athero-susceptible lesser curvature exhibits higher expression and activity of PKCζ, the expression of p53, and the incidence of p53-dependent endothelial apoptosis [408].

In addition to shear stress, the endothelium in vivo is also impacted by cyclic mechanical stretch in a magnitude-dependent manner. Physiological levels of cyclic stretch (5–10% strain) are reported to protect endothelial survival, while excess stretch (15–20% strain) leads to a compromised endothelial barrier and cell apoptosis [408,409]. Importantly, such mechanosensitive apoptosis is associated with enhanced transcription of genes involved in endothelial inflammation, including IL-6, IL-8, MCP-1, ICAM-1 [410], and TGFβ [411]. Several intracellular signaling pathways have been shown to modulate stretch-dependent endothelial apoptosis. The PI3K-AKT cascade is considered to be a well-established pathway to sustain endothelial survival under mechanical stretch [412,413,414], while the production of ROS and activation of stress-responsive p38 MAPK or JNK is pro-apoptotic upon pathological stretch [411]. Recently, Zhuang et al. reported that cyclic stretch increases microparticle production in cultured ECs, with the endothelial microparticle proteomes dependent on the magnitude of stretch (5% versus 15% strain) [415]. Intriguingly, physiological stretch (5% stretch)-induced microparticles suppress EC apoptosis, potentially via activation of Src [415].

A continual endothelium is critical to suppressing atherogenesis and stabilizing established atherosclerotic plaques [415]. Superficial erosion of plaques, as a consequence of EC apoptosis, has been recognized as a potential cellular event triggering atherothrombosis. This mechanism involves neutrophil and NETosis, which will be discussed in a later section. A plethora of biochemical and biophysical pro-apoptotic stimuli, as discussed above, stress the endothelium of atherosclerosis lesions. However, a challenging question remains: whether the prevention of EC apoptosis constitutes a viable therapeutic invention to treat atherosclerosis. Several important and clinically approved anti-atherosclerotic therapies, including statins, proprotein convertase subtilisin/kexin type 9 (PCSK9) inhibition, and SGLT2 inhibitors, have been reported to protect against endothelial apoptosis [416,417,418,419,420,421]. However, given the fact that these medicines have pleiotropic benefits over endothelial functions, the role of endothelial apoptosis in the anti-atherosclerotic benefits are difficult to dissect. In fact, evidence from animal models shows that the genetic ablation of EC apoptotic machinery is largely inconclusive. For example, endothelial overexpression of the Fas ligand in hyperlipidemic *Apoe^−/−^* mice retards atherogenesis and is associated with dampened endothelial inflammation and reduced T cell and monocyte infiltration, but is indistinguishable from endothelial apoptosis, as assessed by an in situ TUNEL assay [422]. Further, global caspase-3 KO promotes, instead of inhibits, the growth and necrosis of atherosclerotic lesions in *Apoe^−/−^* mice without alteration of the plasma levels of cholesterol or triglyceride. Though EC apoptosis was not examined in the lesions, this study indicates that the prevention of cell apoptosis may not constitute an effective strategy to treat atherosclerosis. This can likely be attributed to the accumulated dysfunctional ECs upon failure in apoptosis execution [423]. Moreover, two alternative possibilities exist, either non-death functions of the apoptosis machinery [424,425] or the switch to other modes of cell death when the apoptosis cascade is inhibited. For the former, both caspase-3 and caspase-9 have a direct, non-death-inducing role in modulating endothelial barrier functions [424,425]. For the latter, pharmacological or genetic inactivation of caspase was reported to shunt the cells under lethal inflammatory stress into necroptosis, a mode of regulated cell death mediated by the formation of a necrosome, a death-inducing protein complex comprising receptor-interacting protein kinase (RIPK) and mixed-lineage kinase domain-like protein (MLKL). Notably, this apoptosis-to-necroptosis switch has been documented for ECs. For instance, induced deletion of endothelial caspase-8 in 6-week-old mice leads to diminished EC apoptosis and fatal hemorrhagic lesions in the small intestine [426]. Importantly, caspase-8-independent EC death and small intestinal hemorrhage are abolished by the genetic inactivation of necroptotic executioner MLKL [426]. We will discuss necroptosis in the following sections.

### 3.4. Necroptosis

Vulnerable atherosclerotic lesions are characterized by a large necrotic core, which comprises debris of dead cells and cholesterol crystals. The necrotic core is surrounded by layers of SMCs and ECM, termed the fibrous cap. The integrity and thickness of the fibrous cap are critical determinants of plaque stability. Notably, the fibrous cap is also covered by ECs, and endothelial denudation is considered one important contributory factor to atherothrombosis. Recent evidence has suggested that a majority of cell death events in atherosclerotic lesions are due to regulated necrosis. Regulated necrosis has three main modes: necroptosis, pyroptosis, and ferroptosis. These three modes of regulated cell death are triggered by different external stress signals, and have distinct signaling cascades and cellular outcomes. Their relevance to the endothelium in the context of atherosclerosis is the topic of the following sections.

Necroptosis is initiated by extracellular death signals, including TNFα, TNF-related apoptosis-inducing ligand (TRAIL), toll-like receptor (TLR) ligands, interferons, and viruses, which stimulate the formation of a death-inducing necrosome complex containing receptor-interacting serine/threonine-protein kinase 1 (RIPK1), RIPK3, and MLKL [427]. The activation of RIPK1 results in autophosphorylation and interaction with RIPK3, leading to RIPK3 oligomerization and necrosome formation [428,429]. Within the necrosome, activated RIPK3 recruits and phosphorylates MLKL, leading to its oligomerization. The phosphorylated MLKL oligomers interact with plasma membrane phospholipids to form pores [430], leading to membrane permeabilization, release of DAMPs, and necroptotic cell death [429]. Notably, necroptotic cells feature robust release of DAMPs and cytokines, in turn aggravating tissue inflammation [429]. Necroptotic inflammation can also be enhanced by crosstalk with pyroptosis, another form of regulated necrosis. For instance, RIPK3 activates the NLRP3 (NOD-like receptor family, pyrin domain-containing 3) inflammasome, which activates caspase-1 to cleave IL-1β into their mature forms [431]. Inflammasome activation is a key step of pyroptosis, which will be discussed later.

Human atherosclerotic plaques are associated with a noticeable increase in the expression of necroptosis mediators RIPK3 and MLKL, at both the mRNA and protein level [432], with their expression being even higher in individuals with unstable plaques compared to those with stable ones [432]. In hyperlipidemic *Ldlr^−/−^* mice, advanced plaques show a greater presence of RIPK3, primarily in macrophages [433]. When RIPK3 is absent, the development of advanced atherosclerotic lesions is reduced in *Apoe^−/−^* and *Ldlr^−/−^* mice, without affecting early atherogenesis [433]. Decisively, bone marrow transplantation experiments demonstrate that the anti-atherosclerotic effects of RIPK3 KO are mediated by bone marrow-derived cells [433]. Moreover, suppression of MLKL using antisense oligonucleotides or genetic deletion diminishes the necrotic cores of advanced plaques of *Apoe^−/−^* mice without impacting the total atherosclerotic plaque burden [434]. Interestingly, although cell death and necrotic cores are reduced in MLKL-deficient advanced lesions, the lesion lipid content is increased, which can mainly be attributed to exacerbated lipid accumulation in macrophage foam cells [434]. While RIPK1 is also predominantly found in macrophages within human carotid lesions, its deletion in LysM+ myeloid cells potentiates, instead of attenuates, macrophage necroptosis and necrotic core formation in atherosclerotic lesions in *Apoe^−/−^* mice [435]. This complex, regulatory role of RIPK1 in macrophage death can be at least partially attributed to its transactivation of the NF-κB survival pathway [435]. However, a comparative study using mice with different deficiencies in a multitude of necroptosis-related inflammatory disease models, ranging from systemic inflammation sepsis to localized ischemia-reperfusion injury of the kidney, showed that MLKL deficiency provides limited or no protection against adverse outcomes, whereas *Ripk1^D138N/D138N^* (a catalytically inactive mutant) and *Ripk3^−/−^* mice were protected [436].

While strong evidence supports the functional importance of macrophage necroptosis in atherosclerosis, the roles of EC necroptosis in the context of atherosclerosis remain elusive. Recently, Colijn et al. performed an important study to compare the effects of myeloid-, EC-, and SMC-specific KO of RIPK3 on atherosclerosis in *Apoe^−/−^* mice [437]. Mice with RIPK3-deficient ECs exhibit an increase in lesion burden to a similar extent as mice with RIPK3-deficient myeloid cells [437]. Mechanistically, RIPK3 deficiency in ECs enhances the expression of inflammatory mediators, including E-selectin and MCP-1, but had an inconclusive role in necroptotic cell death, as aortas with endothelial deletion of RIPK3 do not show significant changes in total or phosphorylated MLKL [437]. In the same regard, Karunakaran et al. reported that endothelial RIPK1 functions as a central driver of vascular inflammation in atherogenesis, instead of a regulator of cell survival [438]. In HUVECs, shRNA against RIPK1 reduces inflammatory gene expression (IL-1β, E-selectin, and ICAM-1) and monocyte attachment via promoting nuclear translocation of the NF-κB p65 subunit [438]. It must be noted that EC necroptosis has been investigated widely in microvasculature associated with a variety of diseases, including cardiac transplantation [439,440], lung injury [441,442], and tumor metastasis [443,444,445]. One typical example is that lung microvascular endothelial necroptosis serves as an essential event to facilitate the extravasation of circulating, metastatic tumor cells [443]. This intriguing intercellular crosstalk requires direct contact between tumor cells and ECs. EC expression of TNFR1 and death receptor 6 (DR6) mediates necroptotic cell death and opening of the endothelial barrier to facilitate tumor cell transmigration [443,444,445]. Does this intercellular communication also apply to the transendothelial migration of leukocytes into atherosclerotic lesions? Answers may come from more specific in situ analyses of leukocyte–endothelial contact with lesions of different stages and anatomical sites. With this evidence taken together, although the inhibition of necroptosis machinery, such as RIPK1/3 and MLKL, has demonstrated anti-atherosclerotic benefits in animals, the direct role of EC necroptosis warrants further study.

### 3.5. Pyroptosis

Pyroptosis is initiated through the activation of caspase-1 within a large complex called an inflammasome [332,446]. There are multiple forms of inflammasomes, including NLRP-1, NLRP-3, absent in melanoma 2 (AIM-2), NLR family CARD domain containing 4 (NLRC-4), and pyrin. The NLRP-3 inflammasome is the most widely studied in the vascular system, and consists of NLRP-3, apoptosis-associated speckle-like protein containing CARD (ASC), and procaspase-1 [447], all of which are expressed in the arterial wall [448,449]. Evidence suggests that atherosclerotic plaque components, such as oxLDL and cholesterol crystals, can activate NLRP-3 inflammasomes, partially involving lysosomal rupture and consequent cathepsin release [450,451,452]. Active caspase-1 promotes inflammation by converting pro-IL-1β and pro-IL-18 into bioactive forms, and by cleaving Gasdermin D (GSDMD). The cleaved GSDMD then forms pores in the cell membrane, leading to membrane disruption and the release of inflammatory factors to amplify tissue inflammation [332,446]. In addition to the canonical pyroptotic cascade, caspase-11 can directly sense and complex with lipopolysaccharide (LPS) in the absence of TLR4 or a canonical inflammasome in the context of sepsis [453]. Subsequently, active caspase-11 cleaves GSDMD, causing pore formation and pyroptotic cell death. Furthermore, caspase-11-induced GSDMD can also activate the NLRP-3 inflammasome, indirectly triggering caspase-1-dependent pyroptosis and the release of IL-1β and IL-18 [453]. However, the role of caspase-11-dependent non-canonical pyroptosis in atherosclerosis remains to be determined. Moreover, targeting different inflammasomes (NLRP-3, NLRP-1, AIM-2, or NLRC-4) has been considered as an anti-atherosclerotic strategy. Small molecule inhibitors selective for NLRP-3, such as MCC950, OLT1177, and CY-09 [454,455], have been shown to reduce IL-1β production and attenuate atherosclerosis progression in preclinical models. For example, van der Heijden et al. reported that intraperitoneal administration of MCC950 significantly ameliorates the development of atherosclerosis in *Apoe^−/−^* mice, and is associated with reduced plasma IL-1β levels and reduced macrophages in the lesions [456]. MCC950-mediated inactivation of the pyroptotic NLRP-3/ASC/caspase-1/GSDMD pathway in mouse aortas was clearly demonstrated in a subsequent study [457]. Importantly, the IL-1β-neutralizing antibody canakinumab has demonstrated considerable clinical benefits, reducing recurrent cardiovascular events by 17% in patients with a previous myocardial infarction (CANTOS trial), independent of the lipid level [119]. Mechanistically, although macrophages are currently recognized as the cell targets of anti-pyroptotic therapy [457,458], emerging evidence has established a role of EC pyroptosis in atherosclerosis. Many pro-atherogenic or pro-inflammatory factors activate the NLRP-3 inflammasome in ECs, including oxLDL, ROS, trimethylamine-N-oxide (TMAO), low shear stress, and nicotine [459,460]. These external stress signals likely converge on the redox-activated NF-κB-NLRP-3 axis [459,460] for the induction of endothelial pyroptosis. In particular, cholesterol crystals present in atherosclerotic plaques can induce NLRP-3 inflammasome activation and IL-1β secretion in cultured mouse carotid ECs [461], suggesting that the endothelial pyroptotic machine contributes to plaque vulnerability. Moreover, nicotine administration exacerbates the burden of atherosclerotic lesions of *Apoe^−/−^* mice fed a high-fat diet, with a functional involvement of ROS-dependent activation of NLRP3 inflammation, in addition to subsequent cytokine production and pyroptotic EC death [462].

While mice with EC-restricted KO of pyroptosis machinery have not been generated to study atherosclerosis, mechanistic insights can still be gained using mouse models with global deficiency of caspase-1 or NLRP-3. For example, Yin et al. reported that caspase-1 KO reduces early atherogenesis in *Apoe^−/−^* mice [463]. *Caspase-1^−/−^; Apoe^−/−^* mice fed a high-fat diet exhibit decreased endothelial activation, including reduced expression of leukocyte adhesion molecules (ICAM-1, VCAM-1, and E-selectin) and reduced secretion of cytokines/chemokines (chemokine (C-C motif) ligand 3 (CCL3), chemokine (C-X-C motif) ligand 2 (CXCL2), and CXCL10). Notably, oxLDL-induced pyroptosis of cultured mouse aortic ECs is abrogated by caspase-1 KO in a SIRT1-dependent manner [463]. Moreover, Zhuang et al. linked disturbed flow with endothelial NLRP3 inflammasome activation in atherosclerotic mice [464]. Pro-atherogenic oscillatory shear stress downregulates KLF2-dependent FoxP1 expression in ECs, in turn derepressing the activity of the NLRP3/caspase-1/IL-1β cascade to promote atherogenesis [464]. Although pyroptotic cell death was not examined in this specific scenario, a subsequent study found that low shear stress (5 dyn/cm^2^) stimulates pyroptosis of HUVECs via suppressing the expression of miR-181b-5p [465], which directly targets the signal transducer and activator of transcription 3 (STAT3) expression to restrain NLRP3 inflammasome activation and cell death [465]. Furthermore, genetic inactivation of GSDMD, the pyroptosis executioner, effectively blocks the development of atherosclerosis in hyperlipidemic mice. Recently, Puylaert et al. demonstrated that global GSDMD KO leads to reduced atherogenesis, which is associated with smaller necrotic cores and increased SMC-to-monocyte ratios in lesions, indicating that pyroptosis inhibition is a potential approach to stabilizing atherosclerotic plaques [466]. However, Puylaert et al. also noted that cleaved, pore-forming GSDMD species are not detected in human carotid lesions, in contrast with widely available evidence on the pyroptotic death of ECs in culture [467]. For now, it is not clear whether nondetectable expression of endothelial GSDMD is a biological observation or a technical issue. More studies are needed to ascertain the status of in vivo pyroptosis of ECs in human atherosclerosis. Notably, disulfiram, an FDA-approved drug, has been identified as a potent small molecule inhibitor of GSDMD pore formation [467], although its effects on atherosclerosis have yet to be tested [467]. Thus, the inhibition of caspase-1, NLRP3, or GSDMD in dyslipidemic and inflammatory contexts prevents endothelial pyroptosis and improves endothelial survival, thereby preserving vascular intima integrity, reducing vascular inflammation, and further attenuating the progression of atherosclerosis.

### 3.6. Ferroptosis

Ferroptosis was first discovered during anti-cancer small chemical screening as a non-apoptotic, MEK- and iron-dependent oxidative cell death process in tumor cells with oncogenic RAS [468]. The same study identified and characterized the first-generation ferroptosis inducers, erastin and Ras-selective lethal 3 (RSL3). Ferroptosis features excessive iron-dependent lipid peroxidation, associated with plasma membrane rupture with shrinkage of the nanopores and mitochondria [339]. The chemical processes of lipid peroxidation are complex and involve the dysregulation of several different pathways. First, iron overload contributes to the generation of hydroxyl radicals via the Fenton reaction to oxidize lipids. Normally, iron is buffered and stored by ferritin as Fe^2+^. Under the condition of iron overload, likely attributed to excessive hemoglobin, iron chloride, heme oxygenase-1 hyperactivation, or transferrin overabundance, the cytosolic labile/free ferrous iron pool grows to induce ferroptosis. In this regard, iron chelators like deferoxamine have anti-ferroptotic effects [469]. Second, enzymatic peroxidation of unsaturated fatty acid in the phospholipid bilayer can be mediated via the Achaete–Scute family BHLH transcription factor 4 (ACSL4)/lysophosphatidylcholine acyltransferase 3 (LPCAT3)/15-LOX cascade. Small chemical inhibitors of 15-LOX, such as PD146176 and ML351, have been used as ferroptosis inhibitors [470]. Third, and best characterized, is the inactivation of glutathione peroxidase (GPX)4 [471]. GPX4 catalyzes the reduction of lipid peroxides in the cellular membrane. Diminished gene expression of GPX4 or depletion of its substrate, glutathione, can permit the accumulation of lipid peroxides and subsequent membrane rupture and cell death. The depletion of glutathione can be caused by inhibition of the Xc-antiporter system, which is responsible for cellular uptake of cysteine in exchange for glutamate. Notably, the Xc-antiporter system is the molecular target of erastin [472], and GPX4 is inhibited by RSL3. In addition, ferrostatin-1, α-tocopherol, or liproxstatin, which directly trap lipid peroxides, can be used to inhibit the execution of ferroptosis.

High iron levels have long been associated with an increased risk of atherosclerotic CVDs [473,474,475]. Martinet et al. suggested that intraplaque hemorrhage, iron deposition, and lipid peroxidation are common pathological features of advanced human atherosclerotic plaque [458]. In experimental animals, iron intake is a cause of atherogenesis [476,477,478,479]. Dietary iron aggravates atherosclerosis, while restricting iron intake via a low-iron diet or iron chelator ameliorates atherogenesis. Recently, Vinchi et al. showed that, in the presence of genetic iron overload caused by a heterozygous mutation of iron exporter Ferroportin C326S, *Apoe^−/−^* mice develop severe spontaneous atherosclerosis in the absence of a high-fat diet, which can be rescued by iron restriction (low-iron diet or iron chelator deferasirox) [477]. The iron-overloaded arterial wall displayed an elevated inflammation profile, including cytokine expression and endothelial dysfunction. In particular, a wide range of endothelial defects were observed, including increased aortic endothelial permeability, elevated expression of ICAM-1, VCAM-1, and E-selectin, reduced eNOS activity, and enhanced oxidative damage (as assessed by the nitrotyrosine content) [476]. Interestingly, iron overload, induced by ferric ammonium citrate, enhanced ROS and apoptosis in cultured human aortic ECs, but ferroptotic markers in dying ECs were not tested in this study [476]. Nevertheless, ferroptosis has been established as a mode of oxLDL-induced EC death [480,481,482]. The nature of endothelial ferroptosis is usually identified using ferroptosis inhibitors (e.g., ferrostatin-1) or molecular markers (iron content, GPX4 level, Solute Carrier Family 7 Member 11 (SLC7A11), the Xc-antiporter protein). For example, Bai et al. showed that intraperitoneal injection of ferrostatin-1 reduces the atherosclerotic lesion burden of *Apoe^−/−^* mice fed a high-fat diet, to a similar extent to oral administration of simvastatin [481]. Mechanistically, ferrostatin-1 treatment reduces serum and aortic iron content, restores the expression of GPX4 and SLC7A11, and reduces serum LDL-cholesterol levels. Notably, lesional endothelial expression of GPX4 appears to be increased with the treatment of ferrostatin-1 [481]. In addition, ferrostatin-1 rescues oxLDL-induced ferroptotic traits (mitochondria size, iron content, expression of GPX4 and SLC7A11) and the death of cultured mouse aortic ECs [481]. Prenyldiphosphate synthase subunit 2 (PDSS2) is a key enzyme involved in the biosynthesis of coenzyme Q10. Yang et al. found that serum PDSS2 levels are decreased in patients with coronary artery disease. Global PDSS2 KO increases the development of atherosclerosis induced by a Western high-fat diet. Overexpression of PDSS2 suppresses the oxLDL-induced ferroptosis in cultured human coronary arterial ECs, correlated with normalized levels of iron, glutathione, and ROS [480]. Important evidence also exists for the functional link between ferroptosis and atherosclerosis. Iron overloaded by FeSO_4_ induces both ferroptosis and apoptosis to promote the calcification of HUVECs, which can be alleviated by the application of ferrostatin and the iron chelator deferoxamine [483]. In addition to manipulation of the iron content, encouraging evidence also supports the therapeutic potential of targeting other ferroptotic pathways. Transgenic overexpression of GPX4 in *Apoe^−/−^* mice effectively suppresses atherogenesis, as well as necrotic core formation, in advanced plaque [483]. In cultured mouse aortic ECs, GPX4 overexpression reduces ROS production, the expression of ICAM-1 and VCAM-1, and monocyte adhesion induced by lysophosphatidylcholine or 7-ketocholesterol. Interestingly, both the necrotic and apoptotic modes of cell death are prevented by GPX4. Additionally, using the 15-LOX inhibitor PD146176 to hinder enzymatic lipid peroxidation leads to reduced plaque progression in atherosclerotic rabbits which already have established plaques [483]. Similarly, the genetic removal of 15-LOX contributes to a lower plaque load in *Apoe^−/−^* mice. Collectively, these investigations strongly imply that iron-dependent lipid peroxidation plays a significant role in the development of atherosclerosis.

### 3.7. NETosis

NETosis is a form of programmed cell death that is specific to neutrophils, involving the release of neutrophil extracellular traps (NETs) [335,484]. NETs are web-like structures composed of DNA, histones, and various antimicrobial proteins. They are released by neutrophils to trap and neutralize pathogens, such as bacteria, fungi, and parasites, in the extracellular space [335,484]. During NETosis, the neutrophil undergoes a sequence of morphological changes, including the decondensation of its nuclear material and the mixing of nuclear and cytoplasmic components. This leads to the rupture of the cell membrane and the release of NETs into the surrounding environment. NETs not only contain neutrophil materials, but also gather circulating elements in the blood, such as tissue factors, fibrin, and other procoagulants. While NETosis is essential for the immune response, excessive or uncontrolled NET formation can be harmful and has been implicated in the pathogenesis of various autoimmune and inflammatory diseases, including atherosclerosis [335,484]. Functionally, NET induces the activation of ECs, SMCs, antigen-presenting cells, and platelets, leading to localized tissue inflammation and thrombosis, thereby promoting atherogenesis and atherothrombosis [484,485,486]. Reciprocally, dysfunctional ECs enhance NETosis, amplifying the damage to neighboring cells and further injuring the arterial endothelial lining [487]. Coronary specimens from patients with acute myocardial infarction demonstrate the presence of NETs in both fresh and lytic, but not organized, thrombi, suggesting that NETosis occurs in the early stage of atherothrombosis [488]. Importantly, circulating NET components, including plasma levels of nucleosome and myeloperoxidase–DNA complexes, predict the extent of coronary stenosis, the number of atherosclerotic coronary vessels, and the occurrence of major adverse cardiac events, suggesting NET as a novel biomarker in atherosclerosis [489].

For our specific focus, NET not only induces endothelial dysfunction, but also results in EC death. Gupta et al. found that stressed ECs, conditioned by phorbol 12-myristate 13-acetate, TNF-α, or thapsigargin (ER stress inducer) enhance the NET formation of co-cultured neutrophils, which is partially dependent on endothelial release of IL-8 [490]. Reciprocally, NET causes EC death, as evidenced by the cellular uptake of SYTOX green, a membrane-impermeable nucleic acid dye. EC death can be prevented by DNase that disrupts the backbone of NETs [490]. Saffarzadeh et al. subsequently showed that extracellular histones are responsible for NET-induced cultured EC death, as assessed by lactate dehydrogenase release [491]. Notably, double KO of endogenous Dnase1 and Dnase1-like 3 causes severe NET clots and consequent lethal blood vessel occlusion in mouse models of chronic neutrophilia or sepsis [492]. While the specific mode of EC death caused by NET remains elusive, NET-associated citrullinated histone 4 has recently been reported to cause necrosis-like “lytic” cell death of cultured SMCs within 10 min [493]. Does this lytic cell death mechanism hold true for ECs within the proximal impact of NET? More studies are needed to answer this. However, apoptosis has been suggested as a mode of EC death in the context of NETosis, as is discussed in the following paragraph.

Peptidylarginine deiminase 4 (PAD4), a key enzyme of NETosis, mediates the citrullination of histones. Histone citrullination leads to the decondensation of chromatin, which is essential for the extrusion of DNA and the formation of NETs. PAD4 also takes part in ROS production and granule mobilization in neutrophils [335,484]. Given its central role in NETosis, research has been focusing on the impacts of PAD4 in atherosclerosis. Genetic inactivation of PAD4 in bone marrow abrogates NETosis, diminishes endothelial permeability, and reduces in situ thrombosis in an *Ldlr^−/−^* mouse model of flow-mediated superficial erosion of intima [484]. Notably, a decreased number of TUNEL+ apoptotic ECs were observed in mice with *Pad4^−/−^* bone marrow transplantation or DNase I treatment [494]. Importantly, neutrophils are found at the sites of superficially eroded human plaques, and NETs are localized near where apoptotic ECs reside [495,496]. Disturbed flow also plays a critical role in neutrophil recruitment to erosion-prone lesions and subsequent EC apoptosis and mural thrombosis in a TLR2-dependent manner [496]. Moreover, delivery of a PAD4 inhibitor GSK484 using a collagen IV-targeting nanoparticle reduces NET accumulation at sites of endothelial denudation, which is correlated with improved endothelial lining of superficially eroded arterial intima [52]. These findings indicate that NETosis-mediated apoptotic endothelial denudation is a critical cellular event that drives the superficial erosion of atherosclerotic lesions and atherothrombosis. Therefore, NETs represent a novel therapeutic target for the treatment and prevention of thrombotic complications of atherosclerosis.

### 3.8. Complexity and Limitations of Targeting EC Death in Atherosclerosis

The development and progression of atherosclerosis are closely related to various cell death modalities, including autophagy, apoptosis, necroptosis, pyroptosis, ferroptosis, and NETosis, as summarized in Figure 9. Cell death contributes to the development of atherosclerosis mainly by loss of endothelial lining, promoting endothelial dysfunction, and inflammatory monocyte recruitment. However, there are few registered trials on ClinicalTrials.gov that primarily evaluate a cell-death-targeting treatment in the context of atherosclerosis, although the mainstream medicines (statin, PCSK9 inhibitors, and SGLT2 inhibitors) have observable effects on vascular cell viability/death. Additionally, as was discussed above, a large collection of inhibitors of specific modes of EC death have been designed and have shown preclinical benefits. Why is there limited clinical translatability? There are several potential considerations. First, multiple modes of cell death can co-exist in the complex pro-atherogenic environment, with a composite stress input of hyperlipidemia, hyperglycemia, oxidative stress, and inflammatory stress. On some occasions, even one stress signal, such as H_2_O_2_ or oxidized LDL, can induce multiple modes of EC death. Moreover, different modes of cell death may compensate for each other. A typical example is that when caspase is inhibited in ECs and other cells, necroptosis is activated. This intricate death network could confound the trial of an inhibitor for a specific mode of death. In future, integrated analysis of two or more related modes of cell death should be compared in one atherosclerosis study. This kind of study will pave the way for a feasible strategy to target EC death. Second, pharmacological inhibitors of the death of ECs may impact the death of SMCs and immune cells in a similar or opposite manner, which would make EC-specific targeting impractical, and the outcomes difficult to predict. Recent advancement in single-cell omics at the transcript and epigenetic levels will allow for comprehensive analysis of the responses of different lesion cell types and their communications to a single death-modulating treatment.

## 4. Summary and Future Directions

In this review, we discussed EC senescence and death, with a particular focus on the context of vascular aging and atherosclerosis. The main points related to EC senescence include (1) morphological, biochemical, mechanical, and metabolic features of senescent EC; (2) molecular mechanisms of EC senescence with different stress signals and related contributions to aging and atherosclerosis; and (3) senolytic treatment for atherosclerosis. The main points regarding EC death include (1) molecular mechanisms of different modes of EC death, including apoptosis, autophagic cell death, necroptosis, pyroptosis, ferroptosis, and NETosis; and (2) the complexity and limitations of cell- death-targeting strategies for atherosclerosis.

Understanding how EC senescence and death influence atherosclerosis can determine which senescence and death modalities are most relevant to the design of effective therapeutics. Since the experimental observation of EC senescence and death in atherosclerosis, several milestones have been achieved in this area. First, EC senescence and death are now recognized as causal factors in endothelial dysfunction and key steps in the development and progression of atherosclerosis. Second, the establishment of pharmacological and genetic models of the manipulation of EC senescence and death has improved our understanding of the role of the endothelium in atherosclerosis. Finally, decades of evidence have clearly established that the adequate protection of ECs from oxidative stress- or inflammation-mediated cellular damage appears to be a promising strategy for maintaining endothelial functions in the context of an established atherosclerotic environment.

Herein, we proposed several potential future research directions that will enable us to gain an integrated understanding of EC senescence and death in atherosclerosis and to design novel therapeutics to treat atherosclerosis and related complications. First, atherosclerosis is a complex pathological process involving multiple cell types, and the senescence and death of EC could have localized and long-distance impacts on neighboring ECs and other cell types. Studies utilizing single-cell and spatial omics to understand intercellular communication between senescent or dying/dead or senescent ECs and other cell types, especially in human atherosclerotic lesions, need to be performed. Second, although senescent ECs are reported to be resistant to apoptosis, the relationship between senescence and other modes of death at the level of single ECs remains unresolved, even in culture. Tracking the fates of senescent ECs challenged with different death signals would be helpful. Third, dissecting the relative contributions of different modes of EC death across different stages of atherosclerosis is also critical for the design of endothelial-targeting therapies.

## Figures and Tables

**Figure 1 ijms-24-15160-f001:**
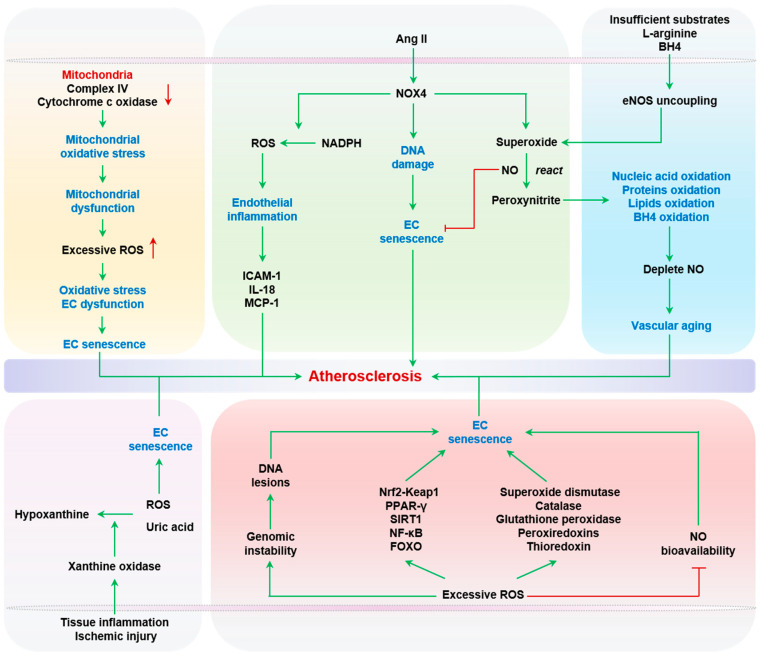
Oxidative stress and EC senescence. Green pointed arrows indicate stimulation; red blunted arrows indicate inhibition.

**Figure 2 ijms-24-15160-f002:**
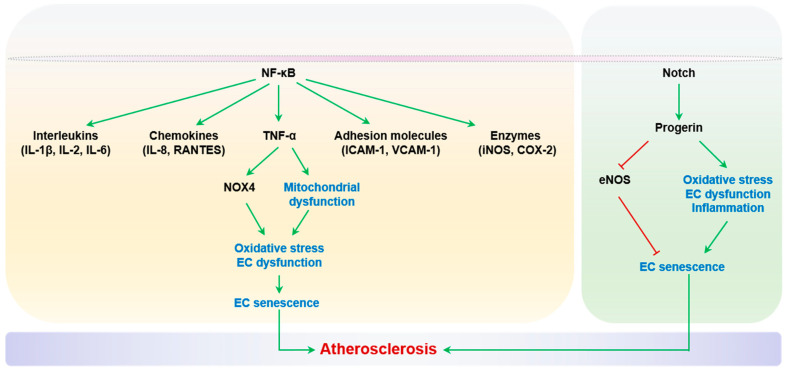
Inflammation and EC senescence. Green pointed arrows indicate stimulation; red blunted arrows indicate inhibition.

**Figure 3 ijms-24-15160-f003:**
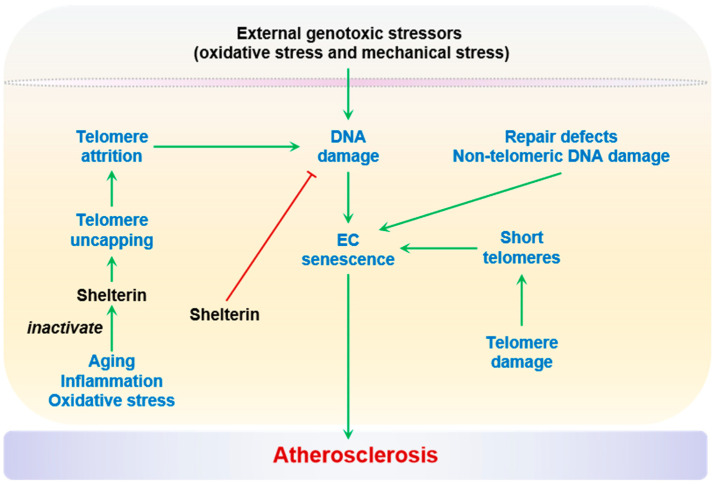
Genomic instability and EC senescence. Green pointed arrows indicate stimulation; red blunted arrows indicate inhibition.

**Figure 4 ijms-24-15160-f004:**
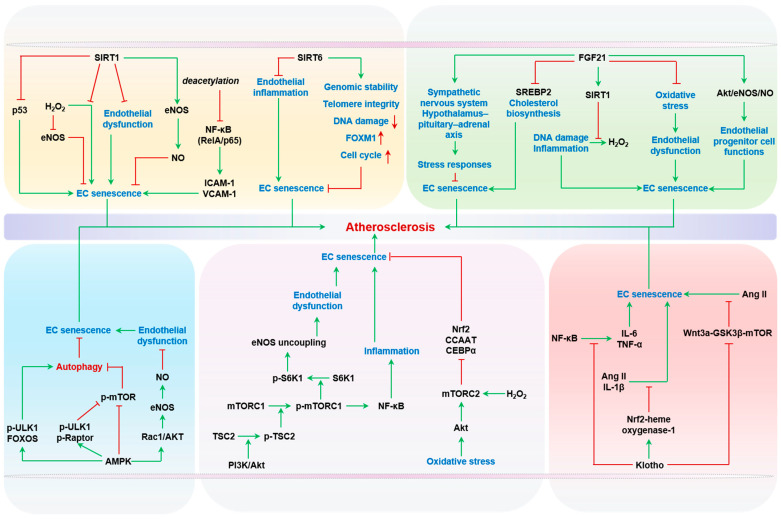
Metabolite-sensing longevity pathways and EC senescence. Green pointed arrows indicate stimulation; red blunted arrows indicate inhibition.

**Figure 5 ijms-24-15160-f005:**
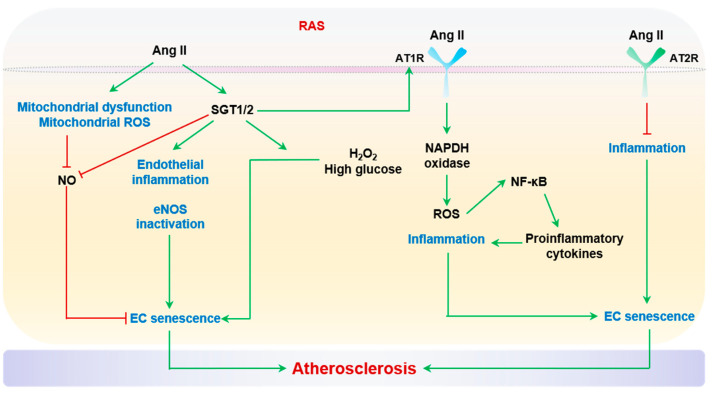
Activated RAS and EC senescence. Green pointed arrows indicate stimulation; red blunted arrows indicate inhibition.

**Figure 6 ijms-24-15160-f006:**
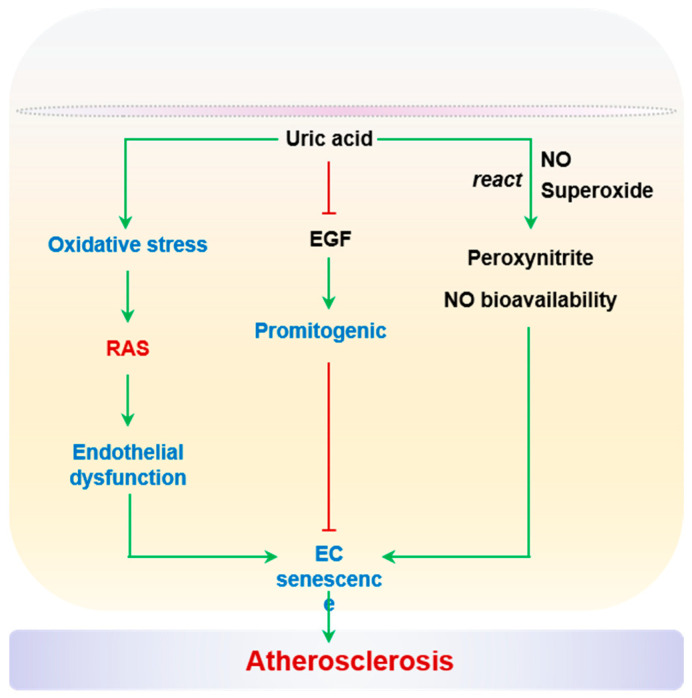
Uric acid and EC senescence. Green pointed arrows indicate stimulation; red blunted arrows indicate inhibition.

**Figure 7 ijms-24-15160-f007:**
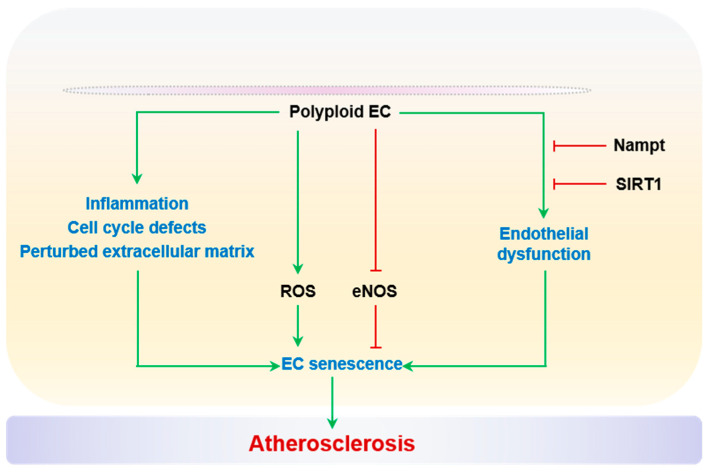
Polyploidization and EC senescence. Green pointed arrows indicate stimulation; red blunted arrows indicate inhibition.

**Figure 8 ijms-24-15160-f008:**
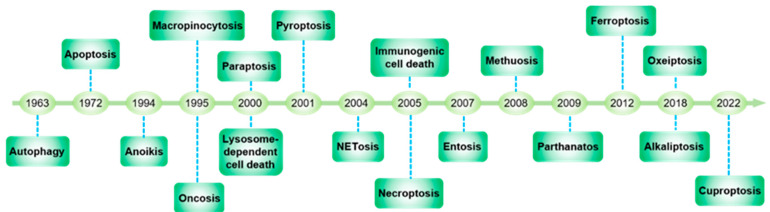
Timeline of cell death research.

**Figure 9 ijms-24-15160-f009:**
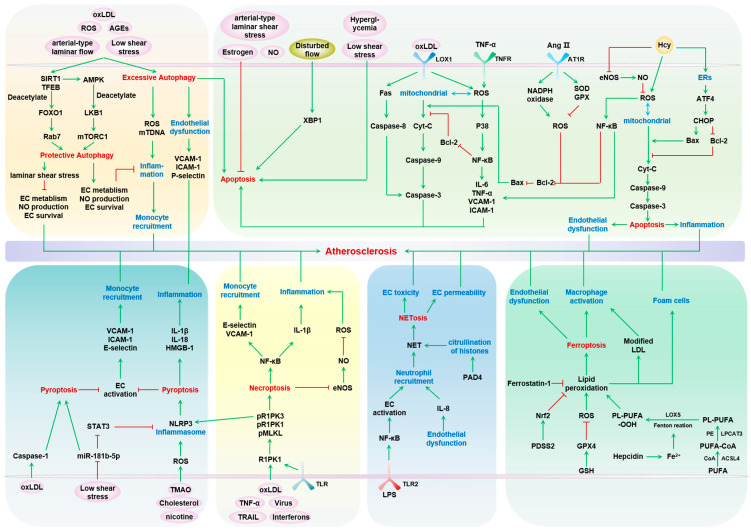
EC death in atherosclerosis. Green pointed arrows indicate stimulation; red blunted arrows indicate inhibition.

**Table 1 ijms-24-15160-t001:** Methods to detect in vivo cellular senescence.

Samples	Indexes	Methods	Advantage	Shortcoming	Ref
Protein (tissues, cells)	SA-β-gal	SA-β-gal test	Gold standard	Traumatically harvested, time-consuming	[36]
	Gal-EM	Transmission electron microscopy	Gold standard	Traumatically harvested, expensive instruments	[37]
	p16, p21, SAHF	RNAscope Immunohistochemistry	High specificity, quantifiable	Operational complexity, time-consuming	[38]
	Lipofuscin	Sudan black B staining	High specificity	Time-consuming	[39]
Serum/Plasma	SASP: IL8/CXCL1/CXCL2/CXCL3 /TNF/IL-1α/IL-1β/ICAM-1/GM-CSF	ELISA Western blot	Mature method, quantifiable	Low specificity	[40]
	TRX80	ELISA	Mature method, quantifiable	Low specificity	[41]
	TNF-α/IL-1β/IL-6	ELISA	Mature method, quantifiable	Large sample sizes are required	[42]
	Endothelial progenitor cells	Flow cytometry	High specificity, quantifiable	Operational complexity	[43]
DNA	Telomere loss	Southern blot qPCR	High specificity, mature method, quantifiable	Operational complexity, time-consuming	[44]
	g (POLG) mutation	DNA sequencing	Mature method, quantifiable	Low specificity, expensive	[45]
	m.3243A > G	DNA sequencing	Mature method, quantifiable	Low specificity, expensive	[46]
	LINE-1 elements	RT-qPCR	Mature method, quantifiable	Low specificity	[47]
	γH2AX	Dual isotope SPECT imaging	Quantifiable	Low specificity, expensive	[48]
Non-coding RNA	miR-183/miR-34a	PCR	Mature method, quantifiable	Low specificity	[49]

Abbreviations: CXCL1/2/3: C-X-C motif chemokine ligand 1/2/3; Gal-EM: galactosidase-EM; GM-CSF: granulocyte monocyte colony-stimulating factor; ICAM-1: intercellular adhesion molecule 1; IL-1α: interleukin 1 alpha; IL-1β: interleukin 1 beta; IL-6/8: interleukin 6/8; p16: cyclin dependent kinase inhibitor 2A; p21: cyclin-dependent kinase inhibitor 1A; POLG: polymerase gamma; SA-β-gal: senescence-associated β-galactosidase; SAHF: senescence-associated heterochromatin aggregation; TNF: tumor necrosis factor; TRX80: thioredoxin 80; γH2AX: gamma histone variant H2AX.

**Table 2 ijms-24-15160-t002:** The main features of cell death.

Cell Death	Morphological Features	Biochemical Features	Key Biomarkers	Hallmarks	Related Signaling Molecules and Pathways	
Autophagy	Formation of double-membraned autolysosomes	Increased lysosomal activity	ATG5, ATG7, DRAM3, TFEB	LC3-II/I, autophagic vesicles, autophagic flow	PI3K-AKT-mTOR, MAPK-ERK1/2 pathway	[326,327]
Apoptosis	Cell membrane blebbing, cell shrinkage, nuclear fragmentation, chromatin condensation, DNA fragmentation, and apoptotic body formation	Activation of caspases, oligonucleosomal DNA fragmentation, modification of nuclear polypeptides	Caspase, p53, Fas, Bcl-2, Bax	Cell morphology, annexin V/PI, mitochondrial membrane potential, cytochrome C	Death receptor, mitochondrial, endoplasmic reticulum pathway, caspase, p53, Bcl-2-mediated pathway	[328,329]
Anoikis	Insufficient cell–matrix interactions	Integrins interact with ECM components to form adhesion complexes	Bcl-2, Bcl-XL, Mcl1, Bax, Bak, Bok, Bid, Bik, Bmf, Noxa, Bad, Bim, Puma	Caspases activation, DNA fragmentation	JNK, ERK, PI3K, PKB/AKT pathway, integrin, caveolin-1	[330,331]
Pyroptosis	Apoptosis-like chromatin condensation and DNA fragmentation in the early stage, karyopyknosis, necrosis-like cell membrane pore formation, cellular swelling, membrane rupture	Dependent on caspase-1 and proinflammatory cytokine releases	Caspase-1, IL-1β, IL-18	Gasdermin D, caspase-4, cell morphology	Caspase-1, NLRP3-mediated pathway	[332,333]
NETosis	Enzymes from granules translocate to the nucleus, chromatin de-condensation, internal membranes break down, cytolysis releases NETs	NETs generation	LPS, GM-CSF, IL8, PAD4, HMGB1, TF, MMP9, PMA, TGFβ, MPO, NE, RAGE, Histone, Anti-DsDNA	Promoting the transcription and translation of NE, MPO, and PAD4	Raf-MEK-ERK, TLR2-fibronectin, caspase-4/5-GSDMD pathway	[334,335]
Necroptosis	Rupture of the cellular membrane, progressively translucent cytoplasm and swelling of organelles, moderate chromatin condensation	Drop in ATP levels, Activation of RIP1, RIP3, MLKL	LEF1, RIP1, RIP3	RIPK1, RIPK3	TNFα, TNFR1, TLR3, TRAIL, FasL, ROS, PKC-MAPK-AP1-mediated pathway	[336,337]
Ferroptosis	Small mitochondria with condensed mitochondrial membrane densities, reduction in or vanishing of mitochondria crista, outer mitochondrial membrane rupture	Inhibition of xCT and reduced GSH, inhibition of GPX4 Iron accumulation, lipid peroxidation	GPX4, Nrf2, LSH, TFR1, xCT	Fe^2+^, ROS, GSH, lipid peroxidation	MVA, HSF1-HSPB1, p62-Keap1-Nrf2 pathway; LSH pathway	[338,339]

Abbreviations: ATG5/7: Autophagy-related 5/7, Bad: BCL-2 associated agonist of cell death, Bak: BCL-2 antagonist/killer 1, Bax: BCL-2-associated X, Bcl-2: BCL-2 apoptosis regulator, Bcl-XL: BCL-2-like 1, Bid: BH3-interacting domain death agonist, Bik: BCL-2-interacting killer, Bim: Bcl-2-like 11, Bmf: Bcl-2 modifying factor, Bok: BCL-2 family apoptosis regulator BOK, DRAM3: transmembrane protein 150B, Fas: Fas cell surface death receptor, GM-CSF: granulocyte–macrophage colony-stimulating factor, GPX4: glutathione peroxidase 4, Histone: hypothetical protein, HMGB1: high mobility group box 1, IL-18/1β/8: interleukin 18/1β/8, LEF1: lymphoid enhancer binding factor 1, LPS: lipopolysaccharide, LSH: solute carrier family 11 member 1, Mcl1: MCL1 apoptosis regulator, BCL-2 family member, MMP9: matrix metallopeptidase 9, MPO: myeloperoxidase, NE: Nelson’s mutant, Noxa: phorbol-12-myristate-13-acetate-induced protein 1, Nrf2: NFE2 like bZIP transcription factor 2, p53: tumor protein p53, PAD4: peptidyl arginine deiminase 4, PMA: Phorbol-12-myristate-13-acetate, Puma: BCL-2 binding component 3, RAGE: advanced glycosylation end-product-specific receptor, RIP1/3: ralA binding protein 1/3, TF: transferrin, TFEB: transcription factor EB, TFR1: transferrin receptor, TGFβ: transforming growth factor beta 1, xCT: solute carrier family 7 member 11.

## Data Availability

Not applicable.

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
