# Peer review of "New Dawn for Atherosclerosis: Vascular Endothelial Cell Senescence and Death"

_ijms, 2023, doi:10.3390/ijms242015160_

Round 1
Reviewer 1 Report
1. Overall, the manuscript is written well to review the molecular mechanisms and functional alterations associated with EC senescence and death in different stages of atherosclerosis.
2. Below are some technical and more subject-related comments and inputs:
a. Since this review paper is principally focused on the molecular mechanisms that underlie senescence, the authors may have to elaborate a little bit about the role of polyploidy, and inclusion of illustrative figures will be good to give an overall idea
b. It will be really informative to include a subheading about “the characteristic features (like morphology, biochemical and metabolic changes) of the senescent endothelial cells” and may discuss furthermore about glutaminolysis.
c. The references for table 2 need to be included
d. Overall, inclusion of precise illustrative figure depicting the molecular mechanistic role of endothelial cell senescence may be informative and add on more value to the review paper
e. Conclusively, I would also suggest adding a section dedicated to the therapeutic strategies (pharmacological and non-pharmacological) for endothelial senescence (including information about senolytics, Navitoclax).
1. Overall, the manuscript is written well to review the molecular mechanisms and functional alterations associated with EC senescence and death in different stages of atherosclerosis.
2. Below are some technical and more subject-related comments and inputs:
a. Since this review paper is principally focused on the molecular mechanisms that underlie senescence, the authors may have to elaborate a little bit about the role of polyploidy, and inclusion of illustrative figures will be good to give an overall idea
b. It will be really informative to include a subheading about “the characteristic features (like morphology, biochemical and metabolic changes) of the senescent endothelial cells” and may discuss furthermore about glutaminolysis.
c. The references for table 2 need to be included
d. Overall, inclusion of precise illustrative figure depicting the molecular mechanistic role of endothelial cell senescence may be informative and add on more value to the review paper
e. Conclusively, I would also suggest adding a section dedicated to the therapeutic strategies (pharmacological and non-pharmacological) for endothelial senescence (including information about senolytics, Navitoclax).
Reviewer 2 Report
The reviewed manuscript is well written and contains a sufficient number of reference sources. I have no issues regarding to the presented manuscript.
Moderate editing of English language required
Reviewer 3 Report
In this review, Lan Bu et al. comprehensively review the role of vascular endothelial cell senescence and death on the progression of atherosclerosis. The authors delve deeply into the intricate molecular mechanisms and functional changes related to these processes. They explore various types of cell death, including autophagy, apoptosis, necroptosis, pyroptosis, ferroptosis, and NETosis, and their role in atherosclerosis. The review emphasizes the effects of oxidative stress, inflammation, genomic instability, and metabolite-sensing longevity pathways on endothelial cell senescence. Also, it discusses the role of the activated renin-angiotensin system and uric acid in these processes. The authors highlight the need for further research on potential therapeutic strategies targeting these pathways.
The article is well-organized and presents information logically and coherently. However, the authors could improve the article by addressing the following points:
- The authors should expand their discussion to include the role of other cell types in atherosclerosis, given the disease's complexity and the involvement of multiple cells and tissue types.
- The potential therapeutic strategies based on the mechanisms discussed in the article should be explored in more depth. A detailed discussion on the progress, challenges, and prospects of developing therapies targeting endothelial cell senescence and death would enhance the article's relevance and applicability.
- The authors should discuss the limitations or challenges in the current research on endothelial cell senescence and death in atherosclerosis.
- The article should provide a clear conclusion summarizing the main points or suggesting future research directions.
- A significant weakness is the absence of diagrams or figures to illustrate complex molecular mechanisms and pathways. Including multiple figures for different sections would significantly enhance the article's clarity and understanding.
Reviewer 4 Report
The authors have reviewed the mechanisms of endothelial cell (EC) senescence and death, including autophagy, apoptosis, necroptosis, pyroptosis, ferroptosis, and NETosis, and how they are related to atherosclerotic progression. Especially insightful is that EC death causes gaps in the endothelium, which attract inflammatory cells to accelerate atherogenesis, and that senescent EC cannot fill in the gaps, even in response to VEGF. This review is extremely well-written, well-organized, coherent, and full of valuable information. Although it is factually dense, the reader is well-rewarded for paying attention. After all the disorganized, disjointed and incoherent submissions I have seen, this restores a sense of sanity to my view of the current scientific enterprise. It is comparable in quality to Annual Reviews, which is a bit of a coup for IJMS. Having said that, there are a few errors that need to be addressed: Most of cells specified in lines 30-31 are not immune cells and, therefore, the sentence should not end with "other immune cells." H2S is hydrogen sulfide, not sulfate. Do the authors mean "chronologic," rather than "chronic" in line 113? Figure 2, which is a masterwork, is difficult to read at 100% view, but, since this is an open access journal, most readers will probably be able to magnify it.
Round 2
Reviewer 3 Report
The authors have done a commendable job in addressing all the concerns. The article looks significantly improved.